# FMRP and MOV10 regulate Dicer1 expression and dendrite development

**Monica C. Lannom**[1], **Joshua Nielsen**[2☯¤], **Aatiqa Nawaz**[1☯], **Temirlan Shilikbay**[1☯], **Stephanie Ceman**[1,3]*

**1** Cell and Developmental Biology, University of Illinois, Urbana, Illinois, United States of America, **2** Integrative Biology, University of Illinois, Urbana, Illinois, United States of America, **3** Neuroscience Program, University of Illinois, Urbana, Illinois, United States of America

☯ These authors contributed equally to this work.
¤ Current address: Dept. of Virology, University of North Carolina, Chapel Hill, North Carolina, United States of America
* sceman@illinois.edu

**Data Availability Statement:** eCLIP data files are available from the NCBI Gene Expression Omnibus (http://www.ncbi.nlm.nih.gov/geo/) under the accession numbers GSE129885 (AGOeCLIP-SEQ).

## Abstract

Fragile X syndrome results from the loss of expression of the Fragile X Mental Retardation Protein (FMRP). FMRP and RNA helicase Moloney Leukemia virus 10 (MOV10) are important Argonaute (AGO) cofactors for miRNA-mediated translation regulation. We previously showed that MOV10 functionally associates with FMRP. Here we quantify the effect of reduced MOV10 and FMRP expression on dendritic morphology. Murine neurons with reduced MOV10 and FMRP phenocopied *Dicer1* KO neurons which exhibit impaired dendritic maturation Hong J (2013), leading us to hypothesize that MOV10 and FMRP regulate DICER expression. In cells and tissues expressing reduced MOV10 or no FMRP, DICER expression was significantly reduced. Moreover, the *Dicer1* mRNA is a Cross-Linking Immunoprecipitation (CLIP) target of FMRP Darnell JC (2011), MOV10 Skariah G (2017) and AGO2 Kenny PJ (2020). MOV10 and FMRP modulate expression of *DICER1* mRNA through its 3'untranslated region (UTR) and introduction of a *DICER1* transgene restores normal neurite outgrowth in the *Mov10* KO neuroblastoma Neuro2A cell line and branching in MOV10 heterozygote neurons. Moreover, we observe a global reduction in AGO2-associated microRNAs isolated from *Fmr1* KO brain. We conclude that the MOV10-FMRP-AGO2 complex regulates DICER expression, revealing a novel mechanism for regulation of miRNA production required for normal neuronal morphology.

## Introduction

Neuronal architecture is affected in many neurodevelopmental disorders. Fragile X syndrome (FXS) is caused by loss of the RNA binding protein (RBP) FMRP [1]. Extensive characterization of FMRP loss in *Drosophila*, mice and humans has led to robust observations revealing the role of FMRP in the development of abnormal dendritic spines [2]. FMRP has also been shown to play an important role in neuronal maturation. FXS patient-derived neurons from induced pluripotent stem cells (iPSCs) and hippocampal neurons from neonatal *Fmr1* knockout (KO) mice and adult *Fmr1* knockdown (KD) exhibit defects in neurite extension and dendritic maturation [3–7].

**Funding:** This study was supported by NIH grant # R01 MH093661 and NSF grant 1855474 to SC. The funders had no role in study design, data collection and analysis, decision to publish, or preparation of the manuscript.

**Competing interests:** The authors have declared that no competing interests exist.

FMRP binds to both the coding region of mRNAs and the 3' UTR [1, 2–9]. It is still poorly understood how loss of a single protein can lead to cognitive impairment although it is known that RBPs seldom act alone, existing in complexes with other RBPs and with the intermediary carrier of information, the mRNA, can enact widespread translational changes depending on their target [2, 10, 11].

FMRP functionally associates with the RNA helicase MOV10 [12], a component of the microRNA (miRNA) pathway and cofactor of Argonaute family members AGO1 and AGO2 [13, 14]. Through Cross-Linking ImmunoPrecipitation (CLIP) experiments, it has been established that FMRP and MOV10 share a common RNA interactome. Unlike FMRP, the *Mov10* knockout (KO) is embryonic lethal [15, 16]; however the *Mov10* heterozygous (Het) mouse has increased anxiety and hyperactivity, which are features shared with Fragile X syndrome [17] and suggest impaired neuronal function [15]. We were thus interested in investigating the consequences of *Mov10 and Fmr1* reduction on dendritic development. Here we show that loss of FMRP and MOV10 leads to impaired dendrite maturation.

MOV10 and FMRP work dynamically to regulate expression of the *Dicer1* mRNA. DICER, a type III endonuclease, generates the final functional miRNA from pre-miRNAs, and is highly regulated at every stage of transcription and translation from primary transcript processing to enzyme activity [18]. DICER associates with AGO2 to facilitate the transfer of the mature miRNA. It is unknown what other RBPs participate in this process. Many groups using different organisms have shown that both FMRP and MOV10 associate with DICER and AGO2 [2, 8, 19, 20]). In addition, a recent paper provides compelling evidence that FMRP binds some miRNAs in regions outside of the seed sequence [20]. This would be an intriguing mechanism for FMRP to recruit specific AGO-miRNA complexes to its bound mRNAs. Here, we provide evidence for local miRNA production through regulation of *DICER1* expression by FMRP and MOV10.

## Materials and methods

### Animals

Experiments were performed on C57BL6/J WT, *Mov10* Het and *Fmr1* KO mice from both sexes (The Jackson Laboratory, Bar Harbor, ME). Animals were kept on a 12/12 hour light/dark cycle with food and water *ad libitum*. All experiments were performed during the light phase (7 AM-7 PM). Animals were treated in accordance and with compliance with Institutional Animal Care and Use Committee (IACUC) guidelines, IACUC protocol 19112.

### Hippocampal neuron culture

*Mov10* heterozygotes were genotyped at postnatal day 0 (P0) using tail samples and DNA was extracted with the KAPA Fast Extract Kit (#KK7103, KAPA Biosystems, Wilmington, MA). After genotyping, mouse hippocampi were dissected and cultured as described [21]. Coverslips were coated overnight with 10 µg/mL poly-L-lysine (#P4704, Sigma, St. Louis, MO) and $10^5$ cells/well were plated for immunofluorescence (IF) in minimum essential medium (MEM) supplemented with 10% fetal bovine serum (FBS). After 24 h, the medium was switched to Neurobasal (NB) medium (#21103049, Gibco, Dublin, Ireland) supplemented with B-27 (#17504–044, Gibco). Half of the media was removed and replaced with fresh NB medium every three days. Neuron culture was performed on at least 3 litters from each genotype.

### Immunofluorescence and microscopy of cultured neurons

Neurons grown on coverslips were fixed in 4% paraformaldehyde for 10 minutes at room temperature. Samples were blocked in 10% normal donkey serum (#017-000-121, Jackson

ImmunoResearch, West Grove, PA) for 30 min at room temperature. MAP2 antibody (1:1000 dilution, # AB5622, RRID: AB_91939, Millipore, Burlington, MA) was incubated overnight at 4˚C. Secondary antibody (Alexa 594 goat anti-rabbit [1:4000, RRID:AB_2307325, Jackson ImmunoResearch, 111-585-144,]) was added for 2 h at room temperature. Coverslips were inverted unto glass slides containing mounting media with 1 μg/mL 4′,6-diamidino-2-pheny-lindole (DAPI). Fluorescence images of DIV14 neurons were obtained with a Zeiss LSM 700 inverted confocal microscope (Zeiss, Oberkochen, Germany) using a 40× and 63× EC Plan-Neufluar 1.30 oil objective respectively. Images were captured with a cooled charge-coupled device (CCD) camera running Zen 2012 software (Zeiss). A total of 10–15 0.2-μM-thick sections were acquired as z-stacks for each neuron imaged.

## Neuronal transfection

P0 brains from *Fmr1* knockout mice *Mov10* heterozygotes mice were harvested for hippocampal neuron culture as above and on DIV2 transfected with pDESTmycDICER, (Addgene plasmid # 19873; RRID:Addgene19873) and empty MYC- or EGFP- expression vector plasmid DNA (Stratagene and Clontech, respectively) using Lipofectamine 2000 (Thermofisher #11668019) for 15 minutes. Half of the transfection media was removed and replaced with Neurobasal (NB) medium (#21103049, Gibco, Dublin, Ireland) supplemented with B-27 (#17504–044, Gibco). After four hours, media was fully replaced, and half of the media was removed and replaced with fresh NB medium every three days. At DIV 7, neurons were prepared for immunofluorescence. Transfection efficiency was measured as percentage of MYC-expressing cells per number of cells plated on coverslip (~50,000) cells and for both conditions, was ~ 0.05%.

## Sholl analysis

Anonymized Sholl analysis of all orders of branches (Total Sholl) was performed using confocal z-stacks of WT, *Mov10* Het, and *Fmr1* KO DIV14 neurons immunostained for MAP2 and imported into ImageJ (Fiji, RRID:SCR_002285). A dendritic complexity analysis, including Sholl analysis, was performed according to the protocol described [22]. Neurite lengths from the soma and soma size area were traced and measured using Image J software and SNT plugin and the data were compiled and analyzed using the Excel program (RRID:SCR_016137, Microsoft, Redmond, WA) and GraphPad Prism (RRID:SCR_002798, San Diego, CA).

## Neurite outgrowth

WT and *Mov10* KO N2A cells were plated in triplicate (density of $1.5 \times 10^4$ cells/well) and incubated at 37˚C in Dulbecco's modified Eagle's medium (DMEM, 10% fetal calf serum). Cells were allowed to differentiate for 48 hours in DMEM (2% FCS) and 20uM Retinoic Acid (Sigma-Aldrich) and imaged under transmitted light using an EVOS cell-imaging microscope (Thermofisher). The images were anonymized and analyzed by an experimenter blinded to the conditions using the Axiovision Image analysis software (Zeiss). 800–1000 differentiated neurons were counted from 10 images per condition.

## Golgi staining

Brain tissue was processed separately in three sets of experiments performed at different times from P14 WT (*n* = 5), *Fmr1* KO (*n* = 5) and *Mov10* Het (*n* = 4). Animals were deeply anesthetized with sodium pentobarbital injection (60 mg/kg, i.p.) and transcardially perfused with 0.9% saline, pH 7.4. The brains were immediately processed for Golgi–Cox analysis using a

standard protocol [23], embedded in celloidin [24] and sectioned in a coronal plane at 175 μm. Spine analysis was conducted as outlined by [24] from somatosensory cortex. Briefly, a 10 μM region was selected from each branch (apical and baslar) and all spines in that region were counted and categorized based on morphology. The same region was subsequently utilized for the dendrite width measurements.

## Western blot

Samples from at least three biological replicates were prepared for immunoblotting after quantification by Bradford assay and suspension in 1× sample buffer, resolved by SDS-PAGE and analyzed by western/immunoblotting. Briefly, membranes were blocked with 10% non-fat dry milk in phosphate-buffered saline (PBS) containing 1% TWEEN-20 for 1 h at room temperature. Primary antibody was applied for 1 h at room temperature or overnight at 4˚C followed by a brief wash in 1% non-fat milk PBS containing 1% TWEEN-20 wash buffer. Horseradish peroxidase (HRP)-conjugated secondary antibody was applied at 1:5000 dilution for 1 h at room temperature and washed 4 × 15 min using wash buffer. The HRP signal was detected using an enhanced chemiluminescent (ECL) substrate and exposed using iBright digital imaging platform. The antibodies used were anti-Dicer (1:100, #sc-393328, RRID:AB_2802128, Santa Cruz Biotechnology, Santa Cruz, CA). anti-eIF5 (RRID:AB_631427, Santa Cruz) at 1:5,000, anti-KIF1A (1ug/ml), (#ab91029, RRID:AB_10862338, Abcam, Cambridge, United Kingdom), 1:1000 anti-MAP1b (# 21633-1-AP, RRID:AB_10793666, Proteintech Group, Rosemont, IL), and HRP-conjugated anti-rabbit and anti-mouse antibodies (RRID: AB_772191, GE Healthcare, Chicago, IL) and Jackson Immunoresearch, (RRID:AB_2338512) respectively. The level of significance and tests performed are described in the Fig legends for each experiment.

## Luciferase reporter assays

Luciferase assay constructs were obtained from Addgene (RRID:SCR_002037, Cambridge, MA). N2A or HEK293T cells were seeded at $5x10^4$ cells into a 24 well plate for 24 hours and transfected with irrelevant or *Fmr1*-specific siRNAs (M-019631-00-0020; D-001810-0X, Dharmacon, Lafayette, CO) using Lipofectamine 2000 (Thermofisher #11668019) for N2A and PEI (Thermofisher #BMS1003-A) for HEK293T for 4 hours. Addition of siRNAs was repeated daily for 72 hours, followed by transfection on Day 4 with luciferase constructs. The procedure was identical for transfection of miRNA mimics (C-310389-05-0002; C-310532-05-0002; C-310427-07-0002, Dharmacon) and pIS1 DICER1 long-mut-miR103/107, (Addgene plasmid # 21652; RRID:Addgene_21652) into N2A cells. For N2A cells, seeding was identical, and transfection of control and luciferase reporter constructs was 24 hours after initial seeding. 1 μg of luciferase (renilla only) containing reporter was transfected along with 10–50 ng of pluc vector (firefly) post knockdown and 24 hours after initial seeding in N2A cells. Luciferase activity was measured in quadruplicate using a dual luciferase reporter assay kit (#E1910, Promega, Madison, WI) on a SynergyTM HT Multi-detection plate reader (Biotek, Winooski, VT) 24 hours post-secondary transfection. Renilla expression from the Dicer "long" 3'UTR construct was normalized to the firefly expression as a control for transfection efficiency. For miRNA mimic and miR deletion and OE experiments, the Dicer "short" 3'UTR was subtracted to account for the effect of MOV10 and FMRP on the luciferase coding sequence itself. Finally, single variate ANOVA was performed to determine if there were any statistically significant differences among the treatment groups versus control followed by a Student's t-test. All measured data are expressed as means +/- SEM.

## eCLIP of P0 WT and *Fmr1* KO brain

P0 Brains from Jax WT C57BL/6 and Fmr1ko mice were sent to Eclipse BioInnovations (San Diego, CA). eCLIP was performed per [25], using anti-AGO2 antibody (EAG009, Eclipse BioInnovations). Briefly, single-end (75nt) sequencing was performed on the HiSeq 4000 platform (Illumina, San Diego, CA). The first 10 nt of each read contains a unique molecular identifier (UMI) which was extracted from each read with UMI tools (version 5.2) and appended to the end of the read name. Next, sequencing adapters were trimmed from the 3' end of each read. Reads were then mapped to a database of mouse repeats using STAR (version 2.6.0c) Reads that mapped to the repeats were removed. The remaining reads were mapped to the mouse genome (mm10) using STAR (version 2.6.0c). PCR duplication removal was performed using UMI tools (version 5.2). CLIP per (version 1.4) was then used to identify clusters within the IP samples, and read density within clusters was compared against the size matched input sample using a custom perl script to identify peaks enriched in the CLIP sample versus the input sample. The significance threshold was -log10(P-value) $\geq$ 3 and a log2 fold change $\geq$ 3.

## miRNAseq of P0 WT and Fmr1 KO brain

University of Illinois Urbana Champaign sequencing center prepared the libraries from three P0 brains from each genotype and sequenced using NovaSeq 6000 and performed FASTQC (version 0.11.8) on individual samples (N = 3 of each genotype). Average per-base read quality scores are over 30 until ~90 bp and no adapter sequences were found indicating those reads are high in quality. The Sequence Length Distribution plot shows a large spike at 22 bp that are the mature miRNA and a smaller spike at 66 which are tRNAs and hairpin/precursor miRNAs. Weighted counts to mature miRNA, hairpin miRNA and tRNAs were generated. Percentages of total reads that mapped to any of these three ranged from 58.6 to 64.8% (S4 Fig). The mapping percentage did not differ between WT and *Fmr1* KO samples. The unmapped reads were discarded while the number of remaining reads (range: 31.5–37.9 million per sample) were kept for further analysis which were then mapped onto each small RNA type.

   A total of 3,429 smRNAs (1,978 mature, 1,234 hairpin and 217 tRNA) were detected. The detection threshold was set at 0.25 cpm (counts per million) in at least 3 samples, which resulted in 2,413 genes being filtered out, leaving 1,016 smRNAs (795 mature, 99 hairpin and 122 tRNA) to be analyzed for differential expression that contain 99.99% of the reads. After filtering, Trimmed Mean of M values (TMM) normalization was performed again (S4 Fig) and normalized log2-based count per million values (logCPM) were calculated using edgeR's cpm () function with prior.count = 2 to help stabilize fold-changes of extremely low expression genes. Differential gene expression (DE) analysis was performed using the edgeR-quasi method (edgeR version 3.30.3). Multiple testing correction was done using the False Discovery Rate method.

## Results

### Both *Mov10* Heterozygote (Het) and *Fmr1* Knockout (KO) cultured hippocampal neurons show abnormal morphology

We cultured hippocampal neurons from *Mov10* Het mice and showed that they have significantly reduced dendritic arborization compared to wild type (WT) [(p < .0001, Fig 1A and 1C) and [15]. Because MOV10 and FMRP bind a common set of mRNAs [15] and colocalize in dendrites [26], it was logical that FMRP would also be required for normal dendritic arborization of hippocampal neurons, as reported by others [3–7, 27].

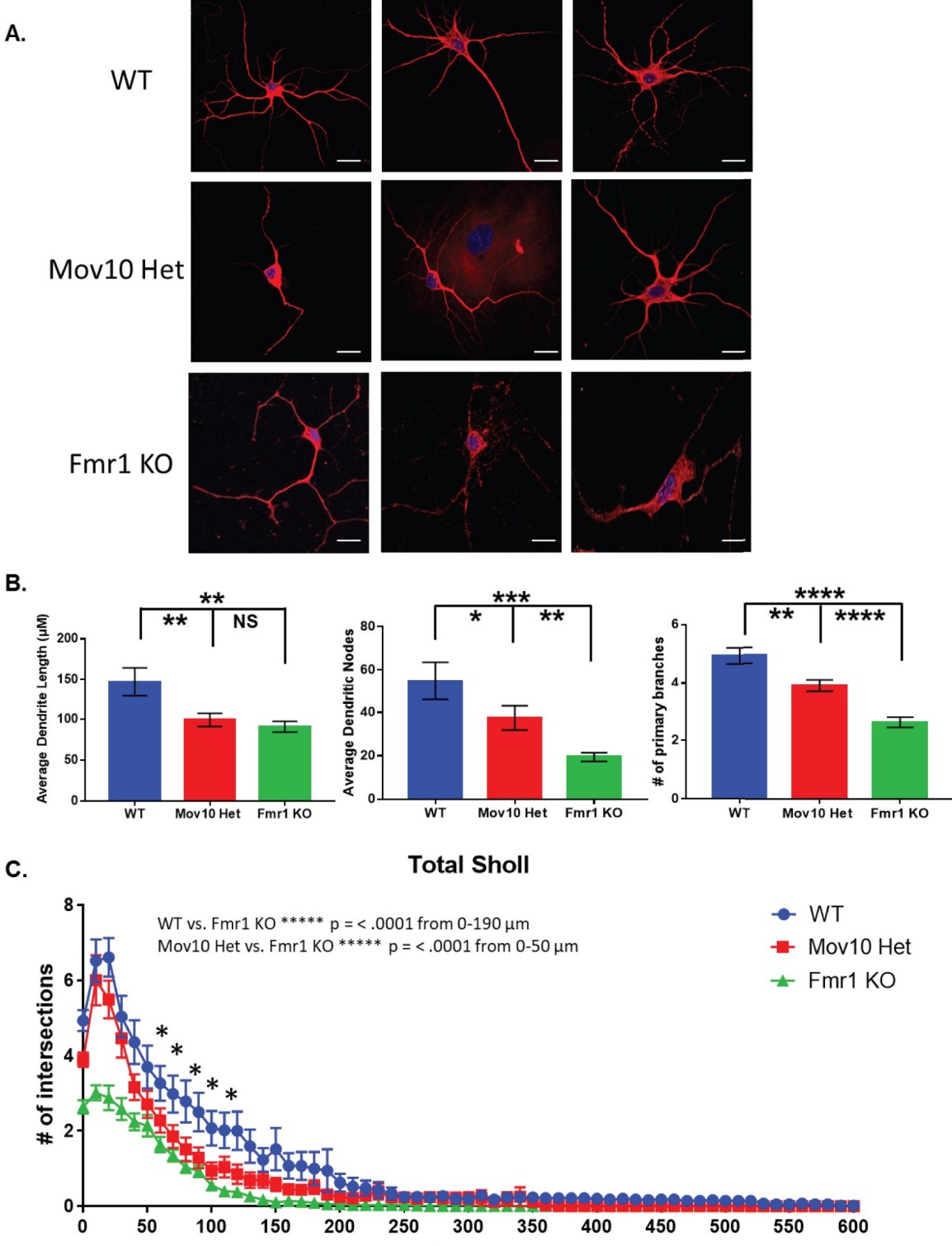

**Fig 1. Mov10 Het hippocampal neurons show an abnormal morphology similar to Fmr1 KO neurons.** (A) MAP2/DAPI immunostaining of hippocampal neurons from DIV14 WT, Mov10 het, and Fmr1 KO neurons. Neurons were prepared from 3 independent litters of each genotype and thus, 3 independent cultures. The total number, N, was compiled from the three biological experiments. (B) Dendritic morphology analysis of average dendritic length, dendritic nodes and primary branches. Confocal z-stacks of MAP2-stained WT, Mov10 het and Fmr1 KO DIV14 neurons were analyzed. (C) Dendritic morphology analysis. Confocal z-stacks of

MAP2-stained WT, Mov10 het and Fmr1 KO DIV14 neurons were analyzed using Sholl. Statistics were calculated using two-way ANOVA followed by Bonferroni multiple comparisons test. Error bars indicate SEM and $^*p < 0.05$; $^{****}$p $< 0.0001$ ($n = 56$ neurons for WT, $n = 94$ neurons for Mov10 Het, n = 58 for Fmr1 KO). Scale bar = 10 μm.

We characterized the neurons by measuring average dendrite length, average dendritic branch points, called nodes, and the number of primary branches. Neurons from the *Mov10* Het and the *Fmr1* KO had significantly shorter dendrites than those of WT; however, dendrite length was not significantly different between the *Mov10* Het and the *Fmr1* KO neurons (p < .001, Fig 1B). In contrast, the average number of dendritic nodes and the number of primary branches in the *Mov10* Het neurons were significantly reduced compared to WT (p < 0.05); furthermore, these same features in *Fmr1* KO neurons were significantly reduced compared to both WT and *Mov10* Het (p < .0001, Fig 1B), suggesting that the reduction in both the number of nodes and the number of primary branches may underlie the results illustrated in Fig 1A. Concordantly, we observed significantly reduced dendritic branching in the *Fmr1* KO neurons compared to WT within 190 micrometers from the cell body. The amount of branching of the *Fmr1* KO neurons was also significantly reduced compared to the *Mov10* Het neurons within 50 micrometers of the cell body, suggesting that complete loss of FMRP was more detrimental to normal dendritic arborization than a 50% reduction in MOV10 (p < .0001, Fig 1C).

Our results agree with independent studies of neurons in *Fmr1* KO mice, which had significant reductions in dendritic complexity, total dendritic length, number of branching points and number of dendritic ends compared to WT neurons in the dentate gyrus [4, 7, 27]. A similar result was observed with FXS human neurons which also exhibited impaired dendritic maturation [7].

It was previously shown that loss of MOV10 and FMRP results in shorter neurites in a murine neuroblastoma cell line (Neuro2A) compared to WT [15]. When we measured the length and width of neurites in *Mov10* KO Neuro2A, we confirmed that in the absence of MOV10, neurites are shorter than in the WT cells independent of the amount of retinoic acid (RA) used to differentiate the Neuro2A cells (p < 0.001, S1A–S1C Fig). Furthermore, we found that the neurites have a larger width in the absence of MOV10 (p < 0.001, S1D Fig).

### Reduced expression of *Mov10* leads to smaller soma size

When we further analyzed our neurons in culture, one individual characteristic stood out: a reduction in soma size between WT and *Mov10* Het (Fig 2). A reduction in cell body size is a feature observed in various X-linked disorders, including human FXS neurons [5] and Rett syndrome [28]. It has also been observed in schizophrenia [28–31]. Accordingly, we quantified the cell soma of the *Mov10* Het neurons compared to WT neurons and discovered a significant reduction in total soma area in *Mov10* Het neurons (p < .05, Fig 2A and 2B). We also measured the soma size of the *Fmr1* ko cultured hippocampal neurons and although it, too, was smaller, it was not significantly different than WT (NS, p < .05, S2A and S2B Fig).

### *Mov10* Het mice have reduced density of immature dendritic spines compared to WT

Because we observed a dendritic phenotype in the hippocampal neurons cultured for 14 days from the three genotypes, we examined dendritic spines in brain sections from animals aged postnatal day 14 (P14) (Fig 3A; [33]). In WT brains, we observed a significantly increased density of the immature spines compared to mature spines (p < .0001, Fig 3B and 3C), which was expected at this particular time point, when rapid pruning is taking place [33, 34]. In contrast, in the *Mov10* Het, the mature spine density was the same as the immature spine density,

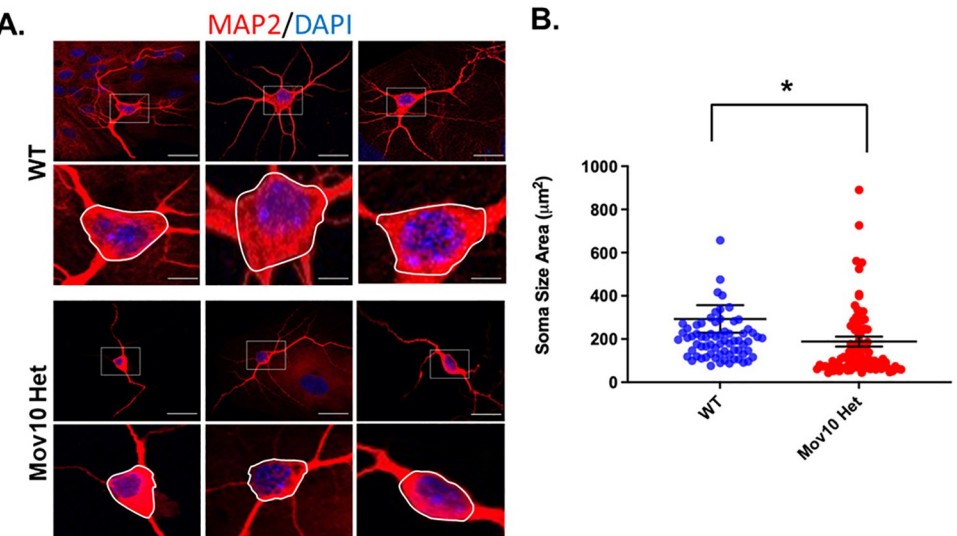

**Fig 2. *Mov10* Het neurons have a reduction in total soma area *in vitro*.** (A) Immunofluorescence microscopy of control (WT) and *Mov10* Het primary hippocampal neural cultures at 14 days *in vitro* (DIV14) showing MAP2 (red) and DAPI [32]. The dashed box indicates the region shown at higher magnification. The area encircled by the white line indicates the region of soma size analysis. (B) Measurements of soma size area in DIV14 primary hippocampal neurons revealed a significant reduction in *Mov10* Het (n = 94) cell body size compared to WT (n = 56). Data are presented as mean ± SEM; *p* values in relation to control (*$p < 0.05$), (Student's *t*-test with Welch's correction). Scale bar: 25 μm. Soma sizes are in S2 Table.

suggesting an increase in the rate of immature spine elimination when MOV10 is reduced (NS, Fig 3B and 3C). Thus, MOV10 appears to stabilize immature spines (Fig 3C). This is a novel role for MOV10 because in *Fmr1 KO* animals, the ratio of immature to mature spines is skewed towards thinner, immature spines in four-week-old and adult mice [24, 33, 35, 36–39]. When we examined the spine density of mature and immature spines at P14 in *Fmr1* ko mice, we observed no difference between WT and *Fmr1* ko mice. We conclude that FMRP and MOV10 participate in dendritic arborization but have distinct roles in spine maturation.

We observed no difference in the width of the apical dendritic branches between WT and *Mov10* Het at P14. However, the widths of the apical oblique and basilar dendritic branches were significantly decreased when MOV10 was reduced (NS, p < .05, p < .0001, Fig 2D). Recent work shows that dendritic widths may be shaped by intracellular transport and forces from the cytoskeleton and the area proportionality accords with a requirement for microtubules to transport materials and nutrients for dendrite tip growth [40]. Thus, reduced MOV10 levels likely perturb dendrite formation because MOV10 binds cytoskeletal mRNAs [15].

The impaired neurite phenotype, reduced soma size, and spine maturation have previously all been found to be associated with impaired miRNA biogenesis [41–44]. miRNAs are 22–26 nucleotides long [45] and are produced upon processing from a longer precursor RNA by the endonuclease DICER [46–51]. Once processed, miRNAs complex with AGO2, forming what is referred to as the RNA induced Silencing Complex (RISC). Since both MOV10 and FMRP are known interactors of AGO2 and are involved in miRNA-associated regulation [12, 26, 52–57], we asked whether miRNA biogenesis could possibly be affected by the loss of MOV10 and FMRP.

## Global reduction of AGO2-associated miRNAs in the absence of FMRP

We were interested in the FMRP dependence of AGO2-association with RNAs, thus, queried miRNA association with AGO2 in WT and *Fmr1* KO P0 brain using enhanced CLIP (eCLIP)

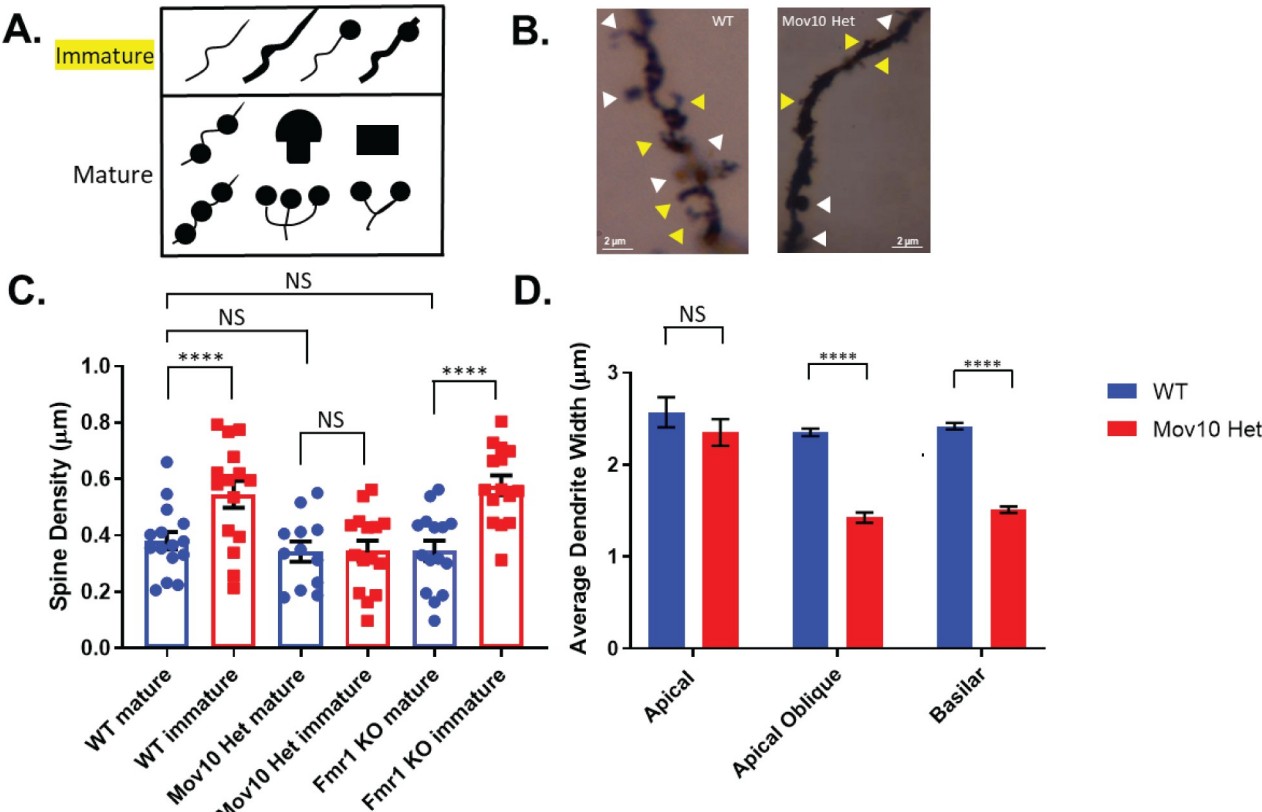

**Fig 3. *Mov10* Het mice have fewer immature spines and wider apical oblique and basilar (secondary and tertiary) branches.** (A) Spine morphology analysis was classified into one of ten different groups, which were further subdivided into immature (yellow) or mature spines (white) based on [34]. (B) Representative images of cortical neurons from WT and *Mov10* Het mice using Golgi staining, scale bar = 2 μm. (C-D) Spine density and average dendrite width were measured per 10 μm dendritic segments of n = 5 (WT animals (8 neurons total) and n = 4 *Mov10* Het animals (8 neurons total) and n = 5 *Fmr1* ko animals. All measured data are expressed as means ± SEM. ****$p < 0.0001$; NS = not significant; $p > 0.05$ (Student's *t*-test with Welch's correction).

[25]. The advantage of eCLIP over traditional cross-linking immunoprecipitation (IP) experiments is that it includes the amount of input RNA in the calculation of the RNA enrichment in the IP [25]. Using this method, we found that the significantly enriched peaks fell within 261 miRNAs (p < .001, Fig 4A, S1 Table), which is a large subset of the 454 miRNAs identified in an earlier study of P13 brain in association with AGO [58]. The miRNAs are highly correlated between *Fmr1* KO and WT but show an average two-fold depletion in the AGO2 IP from *Fmr1* KO compared to WT. Thus, FMRP is required for normal AGO2-miRNA complex formation. Previous work [59] showed that the level of AGO2 protein is the same in WT and *Fmr1* KO mouse brain thus, different AGO2 levels do not explain our result.

Because there is a global reduction in AGO2-associated miRNAs in the absence of FMRP, we hypothesized that one of the miRNA processing proteins could be compromised—either decreased or potentially mislocalized in the absence of FMRP. Examination of the original FMRP brain iCLIP list published by Darnell and colleagues revealed *Dicer1* as a target among the 842 genes identified [60]. *Dicer1* is a MOV10 iCLIP target in brain [15] and *Dicer1* is also a target of AGO2 in mouse brain eCLIP [59]. Thus, *Dicer1* is one of the 29 genes in the intersection of these gene sets (Fig 4B). *DICER1* expression was significantly reduced in *MOV10*-knockdown HEK293 cells [12], suggesting that MOV10 protects both murine *Dicer1* and human *DICER1* mRNAs from AGO2-mediated degradation.

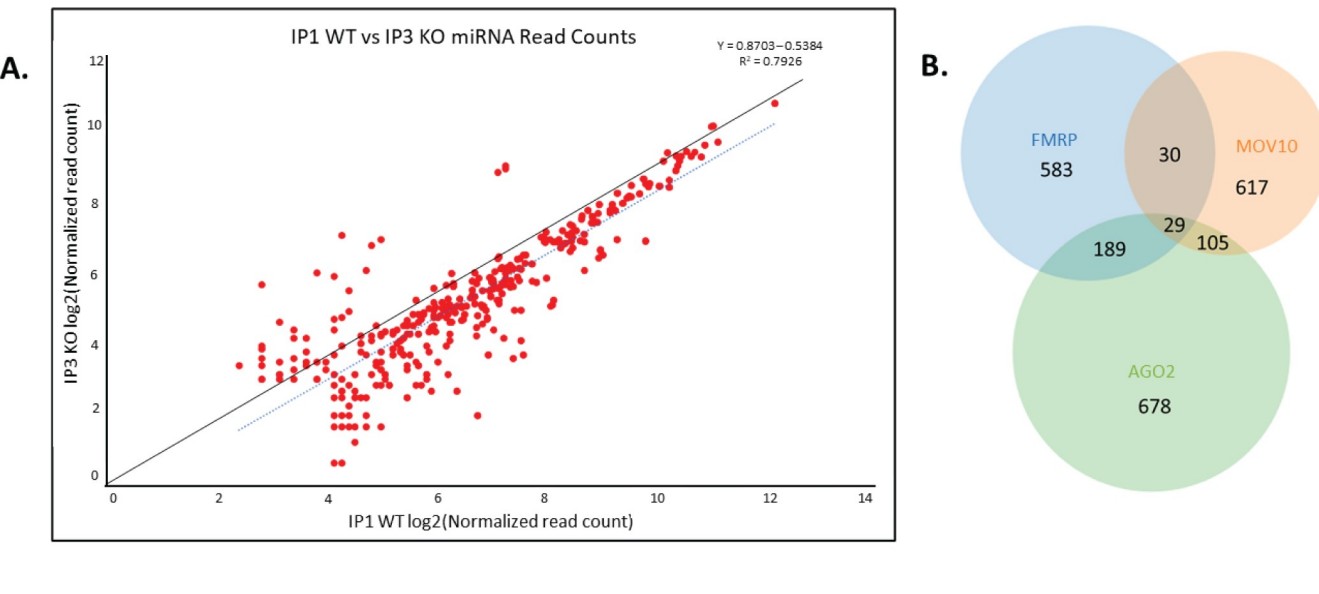

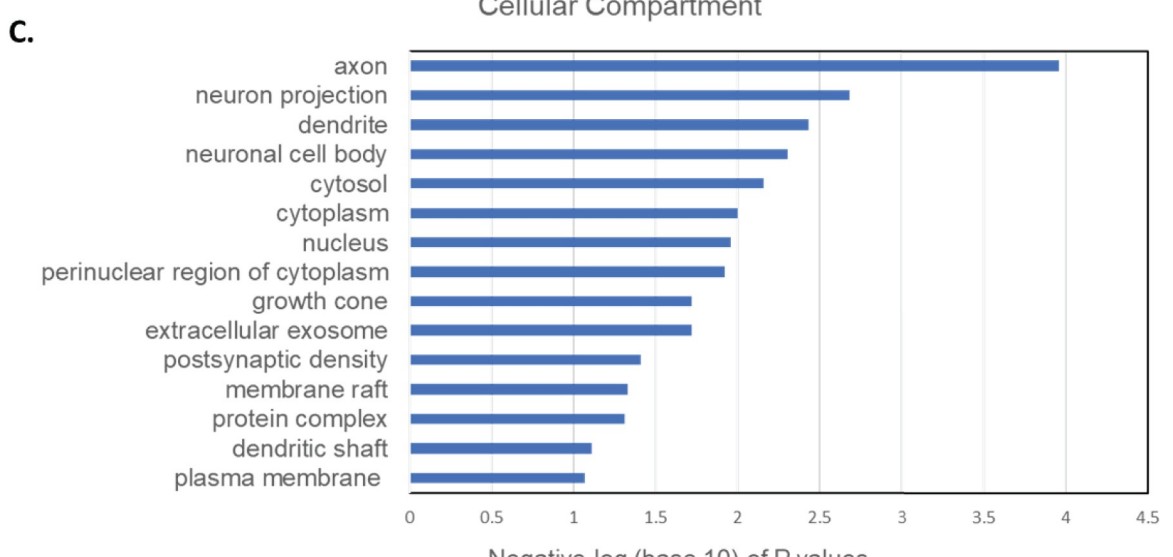

**Fig 4. Global miRNA reduction in brain in the absence of FMRP. (A).** Two-fold less miRNAs associate with AGO2 in the absence of FMRP. Reads per million of WT (X-axis) and Fmr1 KO (Y-axis) AGO2-IPs at each cluster that maps to miRNAs. The IPs had a log2 fold change ≥3 over input and p-value ≤ 0.001. Solid black line = best fit of data. Dashed blue line = actual fit of data. **(B).** Venn diagram showing the overlap between brain-derived iCLIP targets of FMRP [2], MOV10 [15], and AGO2. All three proteins in the brain commonly bound 29 mRNAs (Dicer1 included). **(C).** GO analysis of the shared mRNAs from postnatal brain. *Y axis*: GO terms for Cellular Compartment; *X axis*: negative log (base 10) of the 15 lowest *p* values showing FMRP binds mRNAs encoding proteins involved in neuron projection.

Finally, because of the impaired dendrite phenotype in Fig 1, we used the DAVID Gene Functional Classification tool on the significantly changed AGO2-associated mRNAs between WT and *Fmr1* KO brain from an eCLIP experiment [59] and found the mRNAs encode proteins involved in neuron projection (Fig 4C), as previously shown [15].

## Reduced DICER expression in the absence of FMRP or MOV10

When MOV10 and FMRP bind in the 3'UTR of their mRNA targets, depending on where they bind, the fate of the mRNA changes. When FMRP and MOV10 bind in proximity to each

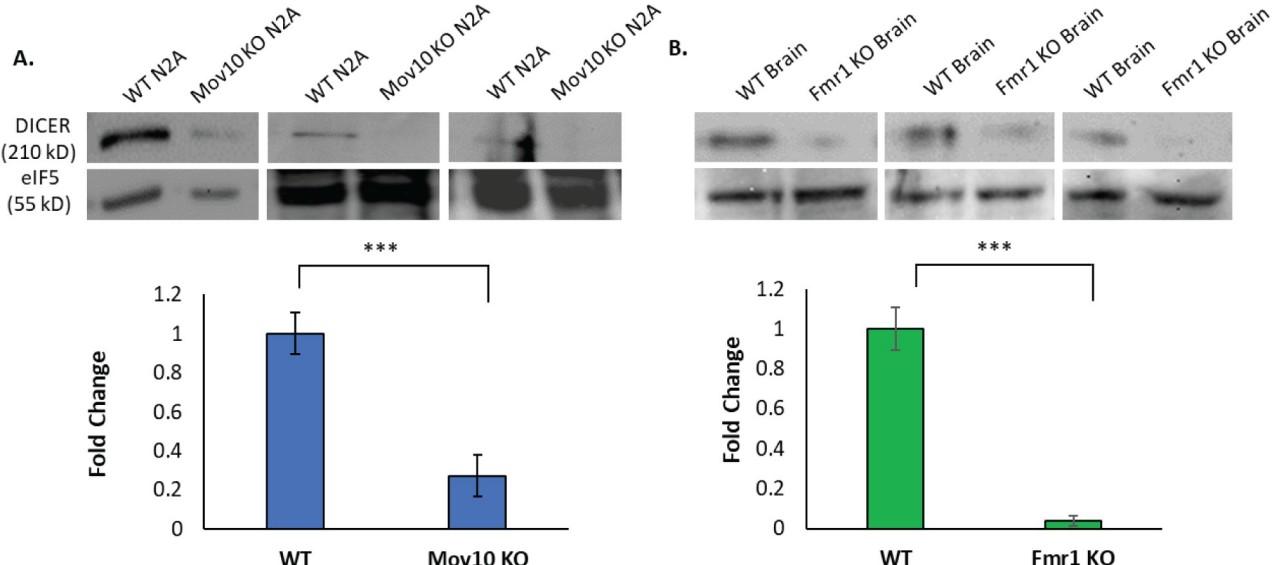

**Fig 5. DICER protein expression is significantly decreased in the absence of MOV10 and FMRP.** (A, B) Three representative images of WT and *Mov10* KO N2A cells (50 μg) and brain extract (50 μg) from P2 C57BL/6 WT and *Fmr1* KO mice immunoblotted for DICER and eIF5 as a loading control. Bar graphs of four and six biological replicates, respectively, are shown below. All measured data are expressed as means ± SEM. ***$p < 0.001$ (Student's *t*-test with Welch's correction).

other on the mRNA, it appears that FMRP binds first and recruits MOV10 to unwind miRNA Recognition Elements (MREs) to facilitate AGO2 association [59]. In contrast, when a G-quadruplex is present in the 3'UTR, FMRP binds it first and binding of FMRP to MOV10 through its KH1 domain stabilizes association of FMRP to the G-quadruplex through its RGG box [59]. If an MRE is present in proximity to the G-quadruplex, association with AGO2 is blocked by the FMRP-MOV10 complex and is temporarily protected from degradation [12, 59]. We hypothesized that the FMRP/MOV10/AGO2 complex regulates *Dicer1* expression by "protecting" the *Dicer1* mRNA, such that in the absence of FMRP and/or MOV10, DICER protein levels would be reduced. When we examined DICER expression in WT and *Mov10* heterozygote brains, we saw no significant difference (NS, p > .05, S3 Fig). Because MOV10 is a helicase, we hypothesized that we would need a complete knockout to see an effect. Thus, we examined DICER expression in cells in which *Mov10* and *Fmr1*, respectively, were knocked out. We observed reduced DICER expression in the *Mov10* Neuro2A KO and in brain extracts from the *Fmr1* KO mouse compared to WT (p < .001, Fig 5A and 5B). It is important to note that the levels of Dicer1 mRNA do not change in N2A cells in the absence of Mov10 [15] (nor in the absence of Fmrp [60, 61]), but *Dicer1* mRNA is highly expressed in brain, similarly to *Mov10* and *Fmr1* mRNA (S3 Fig). Thus, loss of MOV10 or FMRP leads to reduced DICER expression, presumably by allowing access of AGO2 to block translation.

Changes in the levels of DICER in the absence of MOV10 and FMRP could potentially lead to a defect in miRNA biogenesis, thus a significantly decreased pool of mature miRNAs. In fact, other investigators have examined global miRNA levels in *Fmr1* KO brains and found differences from WT [62, 63]. To test this hypothesis, we performed miRNA-seq in P0 WT and *Fmr1* KO brains. Weighted counts to mature miRNA, hairpin miRNA and tRNAs ranged from 58.6 to 64.8% (S4A Fig). The mapped reads overwhelmingly came from mature miRNA ranging from 93.5 to 95.8%, with no obvious difference between WT and *Fmr1* KO brains (S4B Fig). The tRNA reads made up between 4.2 to 6.5% and did not differ between groups (S4B Fig). The precursor/hairpin only accounted for a tiny of fraction of reads, ranging from

0.03 to 0.05%. We had potentially expected to see an increase in hairpin RNA and a decrease in mature in the *Fmr1* KO samples since FMRP regulates *Dicer1* and DICER processes hairpin to mature miRNAs. However, we did not observe any consistent percentage changes between WT and *Fmr1* KO with the exception of two miRNAs, (mmu-miR-144-5p and mmu-miR-3473c). Our results differ from others likely because the methodologies for miRNA measurement differs as well as the ages of the mice [62, 63]. Therefore, the regulation of DICER expression by FMRP and MOV10 does not lead to a global reduction in miRNA levels, despite observing significantly reduced AGO2-miRNA complexes isolated from the *Fmr1* ko brain.

### *Dicer1* 3'UTR regulation by MOV10 and FMRP

From the CLIP-seq data we know that MOV10 and FMRP bind murine *Dicer1* mRNA and human *DICER1* mRNA in the 3'UTR, respectively. To further dissect the role of MOV10 and FMRP on the 3'UTR of the *DICER1* mRNA, we obtained two human *DICER1* 3'UTR luciferase constructs [64]. One contains the entire 3'UTR (referred to as "long") and the second has the 3'UTR truncated (referred to as "short"), essentially removing any possibility for miRNA-mediated regulation (Fig 6A). In the absence of MOV10, luciferase expression of the full length *DICER1* 3'UTR is significantly decreased compared to WT (p < .001 Fig 6C). We also knocked down FMRP in HEK293T cells and observed a similar result (p < .001, Fig 6B and 6D), suggesting that MOV10 and FMRP modulate expression of *DICER1* via the 3'UTR.

   Next we wanted to identify the specific sites in the 3'UTR of *DICER1* through which MOV10 and FMRP exert their effect. To determine this, we re-aligned the binding sites from previously published CLIP-seq experiments [12, 65] to map all sequence sets to the same updated transcriptome. Using these data, we used TargetScan software [66] to determine which miRNA recognition elements (MRE) were the closest to the CLIP sites of MOV10 and FMRP. We then tested several different MREs that according to TargetScan, were highly conserved as potential miRNA binding sites of human *DICER1* mRNA. miRNA mimics for miR-103-3p, miR-195-3p and miR-206 were transfected into WT and *Mov10* KO Neuro2A cells followed by the *DICER1* long 3'UTR luciferase reporter. We found that for the miRNAs tested, MOV10 had the strongest effect on the miR-103-3p site (p < .001, Figs 6E and S5A). Addition of miR-103-3p further suppressed the *DICER1* long luciferase construct in the absence of MOV10, suggesting that MOV10 blocks AGO2+miR-103-3p (Fig 6E, middle columns, green). Moreover, when we introduced a luciferase construct with the miR-103-3p sites deleted, suppression was lifted and expression of the construct in N2A cells was restored (Fig 6E, right columns, red). The fact that loss of the miR 103–3 sites leads to even more expression than WT in the MOV10 knockout suggests that MOV10 facilitates AGO2 association at other now accessible MREs in WT.

### Overexpression of MYC-*Dicer1* rescues impaired neuronal phenotype

To definitively show that it is the loss of DICER expression that is the primary cause of the shortened neurites, we expressed MYC-tagged Human *DICER1* in the *Mov10* KO Neuro2A, *Mov10* HET and Fmr1 KO neurons and observed restored neurite length to WT levels (p < .001, Fig 7A) and improved dendritic arborization within 100 μM of the soma (p < .05, Fig 7B), respectively. Although trending, we did not observe restoration of the dendritic arbor in Fmr1 KO neurons when Dicer1 was OE (Fig 7C). Given that Fmr1 KO neurons had a more severe neuronal phenotype, and that MOV10 and FMRP both regulate many different mRNAs, this result was unsurprising. The data together suggest a mechanism for regulating local DICER expression when MOV10 and FMRP are present.

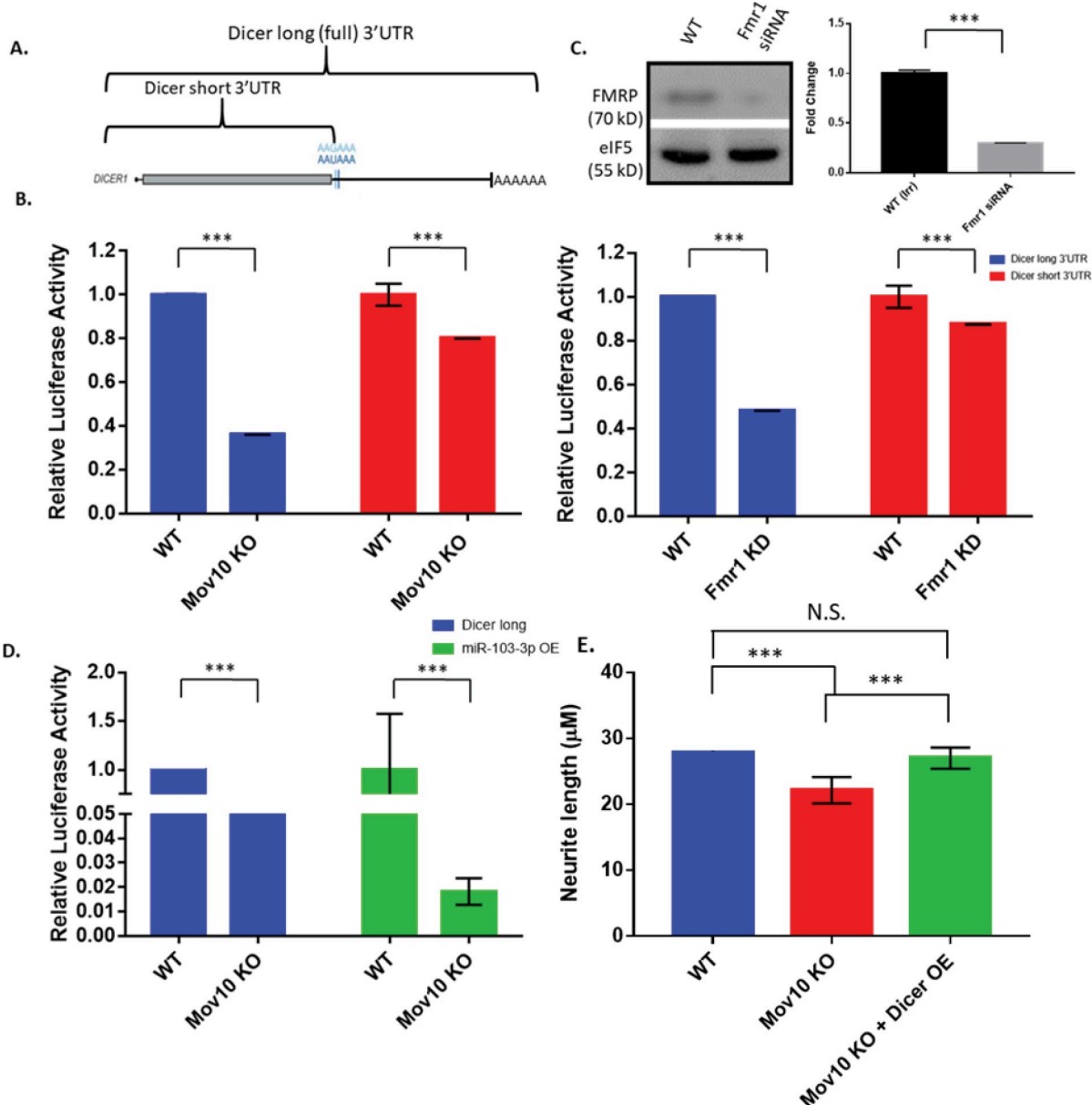

**Fig 6.** MOV10 and FMRP modulate expression of *Dicer1* mRNA via the 3'UTR (A). Schematic illustration of *Dicer1* mRNA with alternative isoforms. Grey boxes show luciferase coding region; black line represents untranslated regions and AAAAAA represents the poly(A) tail. (B). Blot (25ug) and graph showing Hek293T cells were treated with Irr small interfering RNA (*siRNA*) or Fmr1 siRNA (n = 3). (C,D). Effect of Mov10 loss (in N2A cells) and FMRP KD (in Hek293T) on luciferase expression of full-length 3' UTR of *Dicer1* and the shortest *Dicer1* isoform, which was subtracted, as it represents activity independent of miRNA recognition elements. (E) Effect of miR-103-3p overexpression on *Dicer* long reporter and deletion of the miR-103-3p sites from the *Dicer* long construct in the absence of Mov10. Assays were performed in quadruplicate from three independent experiments. All measured data are expressed as means ± SEM. ***$p < 0.001$ (single variate ANOVA).

## Discussion

Our work reveals a new understanding of how FMRP and MOV10 regulate cobound mRNAs and neuronal development. *Fmr1* KO and *Mov10* Het neurons share the features of reduced dendritic arborization, including reduced dendritic length, number of nodes, number of primary branches and reduced soma size. Reducing expression of either protein in Neuro2A cells

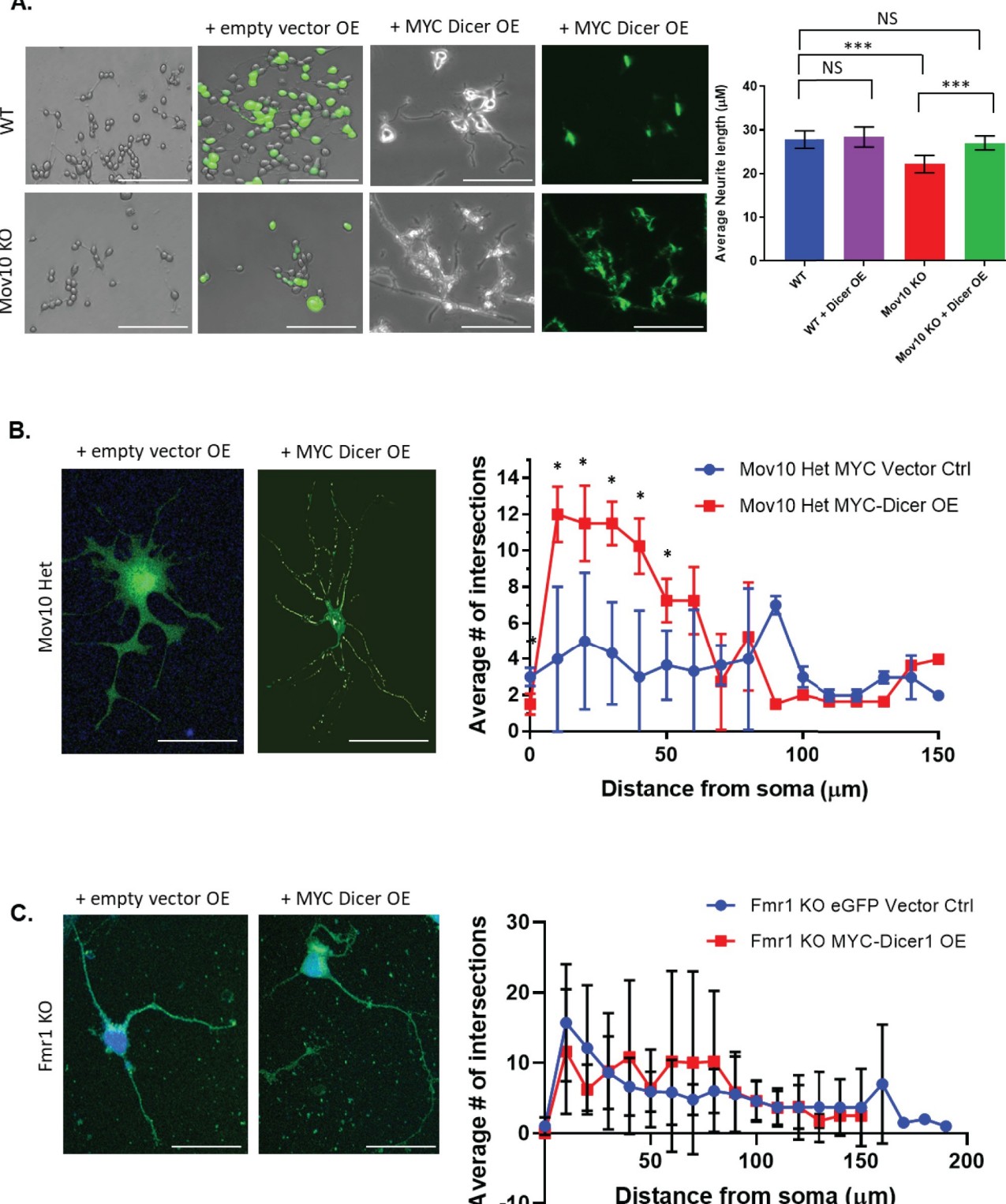

**Fig 7. Overexpression of MYC-*Dicer1* rescues impaired neuronal phenotype.** (A) Brightfield and immunofluorescence images of WT or *Mov10* ko N2A, untransfected or transfected with the empty MYC-vector or the MYC-tagged Human *Dicer1*, as indicated across the top, and stained with the anti-MYC antibody. The average neurite length was measured and shown on the right. Scale bar = 200 μm. Neurite length in micrometers was measured as described in the Methods. All measured data are expressed as means ± SEM. ***$p < 0.001$ (Student's *t*-test with Welch's correction). (B,C). Empty vectors, either MYC or eGFP, and MYC-tagged Human Dicer1 was transfected (over-expressed [OE]) in Mov10 HET hippocampal neurons (B) and *Fmr1 KO*

hippocampal neurons (C) at DIV 2 followed by immunofluorescence for MYC at DIV 7. Sholl statistics were calculated using two-way ANOVA followed by Bonferroni multiple comparisons test. Error bars indicate SEM and $^*p < 0.05$ ($n = 3$ neurons for Mov10 Het empty MYC-vector control, $n = 5$ neurons for Mov10 Het MYC Dicer overexpression, n = 5 for Fmr1 KO MYC Dicer overexpression, n = 5 for eGFP vector control. Bubbles were removed from Mov10 Het OE images for easier viewing. Scale bar = 100 μm.

leads to reduced neurite length and simultaneously reducing both proteins does not lead to a shorter phenotype [15] suggesting that FMRP and MOV10 operate in the same neurite out-growth pathway and dendritic arborization. Although FMRP and MOV10 bind some of the same mRNAs, which includes *Dicer1*, there is also a large number of RNAs that are unique to FMRP and MOV10 and it is likely that misregulation of these mRNAs cause the unique spine features.

Both proteins participate in the miRNA pathway, which plays an important role in normal dendritic arborization. DICER is the primary producer of miRNAs and its mRNA is directly bound by FMRP, MOV10 and AGO2 [15, 59, 60]. We observed significantly reduced levels of AGO2-associated miRNAs in the *Fmr1* knockout brain compared to wild type, initially sug-gesting a global defect in miRNA production in the absence of FMRP; however, that was not the case based on miRNA-seq of both WT and *Fmr1* ko brains, showing that the global miRNA levels were unchanged. To explain our AGO2-eCLIP results, we propose that FMRP participates in loading AGO2 with miRNAs (Fig 8A and 8B). In fact, it was recently shown that FMRP is able to bind miRNAs in regions outside of the seed sequence [20] and miRNAs are present in the FMRP CLIP lists [8].

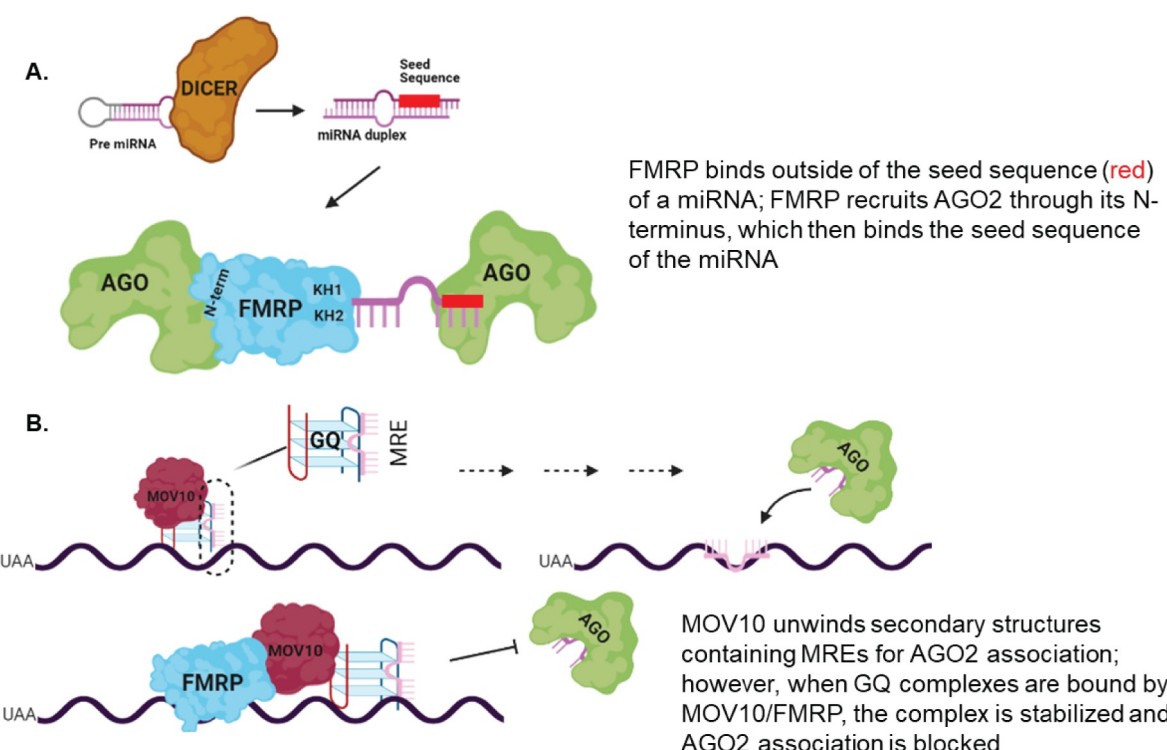

**Fig 8. MOV10 and FMRP interact with AGO2 to regulate *Dicer1* mRNA expression.** (A). Part I of model showing the recruitment of AGO to a miRNA following interaction with FMRP (B). Part II of model showing fate of Dicer1 mRNA bound by MOV10 and FMRP. Binding of both FMRP and MOV10 in proximity of MRE blocks association with AGO2 [12]. In the absence of MOV10 and FMRP, AGO2 is free to bind the MRE and translational regulation takes place. Pink line indicates MRE.

We observed reduced DICER expression in the absence of MOV10 and FMRP in cell lines and brain, respectively, although since DICER is an enzyme, there must be enough present to produce normal levels of miRNAs in brain. Thus, we propose that the role of FMRP and MOV10 on *Dicer* translation affects local expression of DICER in dendrites. In fact, local activation of DICER activity in neurons has been demonstrated before [67]. Dynamically altering local production of miRNAs by changing expression levels of DICER is one way in which neurons could respond to a wide range of temporal and environmental signals [68–70]. The DICER protein itself, along with FMRP and MOV10, is found in neuronal granules and thus can be rapidly dendritically and synaptically localized [71–74]. In fact, neuronal granules typically consist of one mRNA and a large composition of RNA binding proteins [71], suggesting a mechanism in place for rapid control of miRNA synthesis upon signal stimulation.

We hypothesize that FMRP, like most RNA binding proteins, has many different functions in the cell, based on its phosphorylation state, its binding partners and its location in the cell. We might envision that in the cell body, DICER translation is facilitated by the association of FMRP and MOV10 with its mRNA. When the DICER-AGO-MOV10-FMRP complex is transported in the dendrite to the synapse, it awaits stimulation which activates DICER to produce miRNAs locally, which associate with AGO2 through its association with FMRP. Presumably translation is occurring of FMRP-MOV10 bound mRNAs because this complex blocks AGO association. However, translation must eventually stop. Perhaps methylation of FMRP's RGG box releases the FMRP-MOV10 complex from the 3'UTR of synaptically localized mRNAs and AGO now associates with the mRNAs to block translation.

In addition, there may be many roles for MOV10 in the regulation of miRNA production. A recent study showed that shRNA-mediated knockdown of MOV10 in spermatogonial progenitor cells resulted in a significant decrease in most miRNAs. The authors suggest that MOV10 regulates miRNA biogenesis through nuclear RNA metabolism and splicing control, as levels of the miRNA processing proteins were unchanged [75].

In summary, we describe a novel mechanism that has many potential implications. FMRP and MOV10—by regulating the *Dicer1* mRNA, an indispensable element of the canonical miRNA processing machinery—could alter downstream expression of many genes. It has been long known that there is a global increase in protein translation in the absence of FMRP [76]. Much research has focused on finding a single target of FMRP to develop novel therapeutics for patients with FXS [77]. The work presented here might explain why this granular approach has so far not yielded a single gene whose deficiencies explain all of the features of FXS. Our work elucidates how it is possible that so many different genes can be affected by the loss of *Fmr1* and increases our understanding of the role of miRNA biogenesis elements in maintaining proper neuronal cell homeostasis.

## Conclusions

Our work provides a new understanding of how the microRNA processing pathway is regulated and a novel mechanism for how FMRP loss leads to a global increase in translation. DICER is the primary producer of miRNAs and its mRNA is directly bound by FMRP, MOV10 and AGO2. We observed significantly reduced AGO2-associated microRNAs in the *Fmr1* knockout brain compared to WT. DICER protein is also significantly reduced in both the *Fmr1* knockout brain and in the *Mov10* knockout Neuro2A cells, suggesting that FMRP and MOV10 act to block AGO2-mediated suppression of *Dicer1*. This work describes how the FMRP/MOV10/AGO2 complex regulates *Dicer1* expression and suggests that FMRP facilitates loading AGO2 with miRNAs.

## Supporting information

**S1 Fig. Mov10 KO Neuro2A cells have shorter and wider neurites compared to WT.** (A, B). Brightfield images of N2A WT and Mov10 KO cells. Scale bar = 200 μm. (C). Quantification of neurite length of WT and Mov10 KO in the presence of different concentrations of retinoic acid (RA). Between 800–1000 proliferating and differentiated cells were counted from triplicate experiments, and a total of 10 images were counted per condition. (D). Average neurite width in differentiated WT and Mov10 KO N2A cells (1 mM RA) were measured n = 100–250 (WT and Mov10 KO). All measured data are expressed as means ± SEM. ***$p < 0.001$ (Student's $t$-test with Welch's correction).
(TIF)

**S2 Fig. No change in total soma area *in vitro* between WT and Fmr1 KO neurons.** (A) Immunofluorescence microscopy of control (WT) and Fmr1 KO primary hippocampal neural cultures at 14 days *in vitro* (DIV14) showing MAP2 (red) and DAPI (Jentarra et al., 2010). The dashed box indicates the region shown at higher magnification. (B) Measurements of soma size area in DIV14 primary hippocampal neurons in Fmr1 KO (n = 58) compared to WT (n = 56). Scale bar: 25 μm. Data are presented as mean ± SEM; $p$ values in relation to control (NS = $p > 0.05$), (Student's $t$-test with Welch's correction).
(TIF)

**S3 Fig. DICER protein levels do not change in brain when Mov10 is reduced.** A) Whole P2 WT and Mov10 Het (25 μg) were immunoblotted against DICER with Ponceau S as a loading control in three independent experiments. Error bars represent SD, and p values were obtained by Student's t test with Welch's correction (NS > 0.05).
(TIF)

**S4 Fig. miRNAseq reveals no difference in the level of mature vs. immature microRNAs in the absence of FMRP.** A). Multidimensional scaling on the top 500 most variable genes in brain samples. B). Percentages of total reads in the sequencing report that mapped to any of these three: mature, hairpin and tRNA ranged from 58.6 to 64.8%. The mapping percentage does not differ between WT and FMRKO samples. C). Percentage of mapped reads coming from mature miRNA, hairpin miRNA, and tRNA.
(TIF)

**S5 Fig. Overexpression of miR-195-3p and miR-206 does not significantly decrease Dicer 3'UTR expression in the absence of Mov10.** (A). Effect of Mov10 KO on luciferase Dicer1 3′ UTR and miR-195-3p and miR-206 site overexpression. (B) Screenshot from Integrated Genome Browser (IGB) of Dicer1 3'UTR (running from right to left) with human AGO2 CLIP sites (pink), human MOV10 CLIP sites (green) and human FMRP CLIP sites showing relative locations to the MRES containing sites for miR-195-3p and miR-206 binding. (C). TargetScan screenshot of Dicer1 3'UTR (running from right to left) with red circles showing location of MRE sites whose miRNAs were overexpressed in A. Assays were performed in quadruplicate from three independent experiments. All measured data are expressed as means ± SEM. ***$p < 0.001$ (Student's $t$-test with Welch's correction).
(TIF)

**S1 Table. AGO2 eCLIP miRNAs.**
(XLSX)

**S2 Table. Soma size area in WT, Mov10 Het and Fmr1 KO neurons.**
(XLSX)

**S1 Raw image.**
(TIF)

## Acknowledgments

We thank Dr. Roberto Galvez and Dr. Amogh Belagodu for advice on Golgi staining, spine measurement and interpretation of the data. We thank Dr. Hee Jung Chung and Dr. Sung-Soo Jang for instruction on preparing hippocampal neuron cultures. We thank Adriana Tienda for measuring the spines, Malaak Yeyha and Megan Ringling for neurite measurements, and Dr. Lisa Stubbs for critical reading of this manuscript.

## Author Contributions

**Conceptualization:** Monica C. Lannom, Stephanie Ceman.

**Data curation:** Monica C. Lannom, Joshua Nielsen, Aatiqa Nawaz, Temirlan Shilikbay.

**Funding acquisition:** Stephanie Ceman.

**Investigation:** Monica C. Lannom, Joshua Nielsen, Aatiqa Nawaz, Temirlan Shilikbay.

**Supervision:** Stephanie Ceman.

**Writing – original draft:** Monica C. Lannom.

**Writing – review & editing:** Monica C. Lannom, Aatiqa Nawaz, Temirlan Shilikbay, Stephanie Ceman.

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
