## [Decision Letter · Decision Letter 0]

17 Jun 2021

PONE-D-21-14401

FMRP and MOV10 regulate Dicer1 expression and dendrite development

PLOS ONE

Dear Dr. Ceman,

Thank you for submitting your manuscript to PLOS ONE. After careful consideration, we feel that it has merit but does not fully meet PLOS ONE’s publication criteria as it currently stands. Therefore, we invite you to submit a revised version of the manuscript that addresses the points raised during the review process.

The reviewers asked for a better presentation of your data and the inclusion of some controls with the purpose to increase the robustness of your conclusions.

We look forward to receiving your revised manuscript.

Kind regards,

Barbara Bardoni

Academic Editor

PLOS ONE

Journal Requirements:

2. Thank you for including your ethics statement:  "There is no human research

IBC 4571

Animals were treated in accordance and with compliance with Institutional Animal Care and Use Committee (IACUC) guidelines, IACUC protocol 19112.We used the method of euthanasia described in the "AVMA guidelines on Euthanasia". Briefly, the animals were asphyxiated by pumping CO2 into an enclosed chamber until they ceased moving and stopped breathing. We confirmed death by cervical dislocation. P0-P2 pups were euthanized by decapitation. Anesthesia was not used for decapitation as is common for euthanasia of young mice. Severing occurred with a sterile scalpel. ".   

a.) Please amend your current ethics statement to include the full name of the ethics committee that approved your specific study.

b.) Please amend your current ethics statement to confirm that your named ethics committee specifically approved this study.

For additional information about PLOS ONE submissions requirements for ethics oversight of animal work, please refer to http://journals.plos.org/plosone/s/submission-guidelines#loc-animal-research  

Reviewers' comments:

Reviewer's Responses to Questions

**Comments to the Author**

1. Is the manuscript technically sound, and do the data support the conclusions?

Reviewer #1: Partly

Reviewer #2: Partly

Reviewer #3: Yes

2. Has the statistical analysis been performed appropriately and rigorously? 

Reviewer #1: Yes

Reviewer #2: No

Reviewer #3: I Don't Know

3. Have the authors made all data underlying the findings in their manuscript fully available?

Reviewer #1: Yes

Reviewer #2: Yes

Reviewer #3: No

4. Is the manuscript presented in an intelligible fashion and written in standard English?

Reviewer #1: Yes

Reviewer #2: Yes

Reviewer #3: Yes

5. Review Comments to the Author

Reviewer #1: The study by Lannom et al. revealed the importance of MOV10 and/or FMRP in regulating Dicer expression, Ago2 interaction with miRNAs and dendrite development in neuron. They further demonstrate that both proteins act on the 3’-UTR of the Dicer mRNA, and over-expression of Dicer can reverse and neurite deficits in MOV10 KO neurons. Overall the findings appear solid and potentially interesting, but there are a number of concerns that should be addressed.

My biggest concern is the lack of consistency across different experiments that make it hard to draw some of the conclusions. For example, the authors suggest that Dicer expression is only reduced in FMR1 KO brain but not MOV10 Het KO, but the brains they used were P2 while the spine defects of the MOV10 KO mice in Fig. 2 were observed at P14 (which match the 14 DIV neurons in Fig. 1). Is it possible that MOV10 deficiency in the Het KO reduces Dicer expression at P14, which can potentially explain the spine and dendrite phenotypes? Likewise, Fig. 1 examines dendrite defects in cultured hippocampal neurons while Fig. 2 examines spine defects in cortical neuron in vivo using Golgi stain. It would be much better if they correlate dendrite and spine defects in the same group of neurons (either hippocampal or cortical, in vitro or in vivo). Furthermore, the authors claim that the spine defects in MOV10 KO are different from FMRP KO, but the spines from MOV10 KO are at P14 while the spine defects in FMR1 KO mice are implicated from studies by other labs done at different ages (4-wk old and adult, p.14). Without side-by-side comparison in the same experiments, same age and same group of neurons it Is hard to draw this conclusion. In Fig. 6C the MOV10 KO are N2A cells and FMR1 KD is done in 293T cells. They should not be put together in the same graph for comparison.

Other comments:

(1) Fig. 1: how many independent experiments have been performed?

(2) Fig. 2C: is there statistical significance between genotypes for mature spines and immature spines?

(3) Fig. 4B: does MOV10 KO brain show changes in the amount of Ago2-associated miRNAs compared to WT? And what do the two lines in the plot refer to?

(4) Fig. 6D upper panel: Does the green signal represent MYC-Dicer OE cells? It should be stated clearly in the figure. The authors should also show the images for all three experimental conditions that match the quantification in the lower panel.

(5) Fig. 6D: Top left image of the upper panel indicates WT + Dicer OE but there is no such condition in the histogram of the lower panel. Does the blue bar represent WT + Dicer OE?

(6) Fig. 6E: why the bars are broken?

(7) Fig. 6F: representative images of the MOV10 KO and FMR1 KO neurons with or without Dicer OE should be shown.

(8) Fig. 6F: if Dicer OE can reverse the dendrite phenotypes of MOV10 Het hippocampal neurons, the authors need to examine whether Dicer expression is indeed reduced in the Het neurons compared to WT.

(9) Does OE Dicer restore the reduction of soma size in MOV10 Het neurons?

(10) P.3, second paragraph: the sentence is too long and should be divided into separate sentences.

(11) I think dendrite width is seldom measured. The authors should discuss the significance of a change in dendrite width observed in their study.

(12) P.19: Kennedy et al. 2020 is not found in the reference list

(13) P.23: “suggest a mechanism for regulating local DICER expression when MOV10 and FMRP are present”. I don’t think the authors have provided evidence showing a local dendritic (but not global) reduction of DICER.

(14) P.23: “there is also a large number of RNAs that are unique to FMRP and MOV10 and it is likely that misregulation of these mRNAs cause the unique spine features.” As mentioned in previous comment, it is not appropriate to conclude “unique spine features” from the available data because the findings on MOV10 and FMR1 KO were not done side-by-side and the ages or even populations of neurons examined are different.

Reviewer #2: FMRP and MOV10 are two RNA binding proteins functionally connected and playing a role in miRNA-mediated translation regulation. In this manuscript, the authors aim to demonstrate that FMRP and MOV10 regulate the expression of the DICER protein, an endonuclease responsible for miRNA maturation, through the miRNA mediated regulation of its mRNA. This regulation would be important for proper neuronal architecture. In this purpose, the authors used a combination of model impaired in MOV10 or FMRP expression to analyze dendrites and spines morphology, miRNA expression and their association with AGO2 and the expression of the endogenous DICER protein or a reporter protein produced from a construct presenting the DICER 3’UTR region.

The question raised in this study is undoubtedly interesting. However, several inaccuracies or inconsistencies make it difficult to properly appreciate the present work. Moreover, to be fully convincing, the experimental data presented here should be reinforced by additional controls and complementary experiments.

In the present manuscript, the authors first showed an altered dendritic arborization in cultured hippocampal neurons from heterozygote Mov10 mice or Fmr1-KO mice (Figure 1, Figure 2) and in Mov10 Het brain sections (Figure 2), confirmed previously published improper neurite development in the murine neuroblastoma cell line Neuro2A Mov10-KO (Sup Figure 1), observed a perturbed mature/immature spines ratio in brain sections from 14 PND Mov10 Het mice compare to WT (Figure 2) and a smaller soma in cultured hippocampal neurons from heterozygote Mov10 mice compare to WT but not in Fmr1-KO (Figure 3 and Sup Figure 2).

Major points:

- Figure 1 A: images and in particular the morphology of the selected Mov10 Het neuron do not reflect the quantitative characterization presented in B. At least 3 images of representative neurons from each condition should be presented.

- Figure1B-E: Statistical tests should be detailed for each experiments. In particular, the number of different neuronal cultures should be specified.

- Figure 3B and Supplemental Fig 2: the number of different neuronal cultures used for statistical analysis should be specified. Could the individual measurements of soma size area be provided?

Next, to connect these morphological alterations to molecular mechanisms, the authors hypothesized that miRNA biogenesis may be affected by the loss of Mov10 and FMRP. They present results showing a two-fold in miRNA association with Ago2 (Figure 4) in Fmr1-KO P0 brain compare to WT. However, no difference could be detected in the global levels of mature or immature miRNA (Sup Figure 4).

Major points:

- Figure 4 A: It seems that there are inconsistencies between the legend of the figure and the text. Is it GO enrichment for FMRP bound mRNA (legend) or differentially Ago2-associated mRNA in WT vs Fmr1-KO (text)? Were they identified by iCLIP or eCLIP? Where is the list available? The text refers to Kenny et al., 2020 and the references section mentions Kenny et al., 2019.

- Figure 4 B/ Sup table 1: legends are not accurate: what are full line, dashed line? How statistical analysis were performed?

- To rule out any indirect effect, the level of the Ago2 protein in WT vs Fmr1-KO P0 brains should be checked.

The Dicer mRNA being a target of FMRP, MOV10 and Ago2, the authors analyzed the expression of the DICER protein and showed that DICER protein expression is decreased in Mov10-KO N2A cells or P2 Fmr1-KO brain extracts compare to WT (Figure 5A-B).

Major points:

To fully support the hypothesis that FMRP and MOV10 modulate the Ago2-miRNA mediated regulation of the Dicer mRNA, additional experiments are required:

- What are the levels of Dicer mRNA in these different models?

- What is the impact of the absence of FMRP on the association of the Dicer mRNA with Mov10? and conversely?

- What is the impact of the absence of FMRP or Mov10 on the association of the Dicer mRNA with Ago2?

Minor point:

- What is the purpose of Figure 5C here?

As MOV10 and FMRP bind murine Dicer mRNA and human DICER mRNA respectively in the 3’UTR, the authors used a DICER1 3’UTR luciferase construct to dissect the potential role of FMRP and MOV10 in the modulation of DICER1 via the 3’UTR. They showed a decreased luciferase activity in the absence of FMRP or MOV10 proteins (Figure 6C).

Major points:

- Figure 6: Both legend and Material and Methods sections poorly explain the luciferase assay and in particular how long and short constructs participate in the calculation of the Relative Luciferase Actvity (Could the authors explain: “Mov10 KO and Fmr1 KD samples were normalized to WT Dicer long 3’UTR and Dicer short 3’UTR values were subtracted for final graph”?)

Minor point:

- Many steps in mRNA biogenesis could affect the reporter protein expression. qPCR to evaluate the expression of the reporter mRNA in the different conditions would strengthen the authors hypothesis.

By CLIP-seq data comparison and miRNA recognition elements (MRE) analysis, the authors determined potential MSE the in close proximity of FMRP and MOV10 binding sites in the DICER 3’UTR (Sup Figure 5).

Minor points:

- Are mouse and human Dicer mRNA 3’UTR (FMRP binding sites, Mov10 binding sites, predicted MSE ) conserved?

- Sup Figure 5: B and C should be harmonized to facilitate the reading.

MOV10-KO negative effect is enhanced miR103-3p over-expression (but not miR-195-3p or miR-206 over-expression) and is abolished by the deletion of the miR103-3p sites on the luciferase construct (Figure 6 E and Sup Figure 5).

MYC-tagged human DICER1 over-expression restores neurites length to WT levels in N2A MOV10-KO and improved dendritic arborization in Mov10 Het cells neurons but not in Fmr1-KO neurons.

Major points:

- Figure 6 D: Incomplete legend

- Figure 6 E: Inconsistencies between text and legend.

- Figure 6 F left:

o Are Mov10 Het cells cultured hippocampal neurons? 7 DIV? 14 DIV? Were analyzed neurons detected for MYC-DICER expression? What is the level of over-expression in these neurons? Is heterologous expression homogenous?

o How does the over-expression of DICER impact dendritic arborization in WT neurons?

o Can the authors comment the difference in the numbers of intersections between WT (Figure 1), Mov10 Het over-expressing negative CT and Mov10 Het over-expressing DICER (Figure 6F)?

- Considering the cooperative model proposed by the authors, what is the effects of FMRP over-expression on the arborization of Mov10 Het neurons?

- Figure 6F right:

o Same questions as above regarding the control of MYC-DICER over-expression in analyzed neurons.

o Why is the negative control for FMR1-KO neurons different from the one used for Mov10 Het neurons?

o Can the authors comment the difference in the numbers of intersections between WT (Figure 1), Fmr1-KO over-expressing negative CT and Fmr1-KO t over-expressing DICER (Figure 6F)?

To conclude, the authors discuss the potential role of FMRP in facilitating the loading of miRNA on AGO2 (Figure 7A). In parallel, they propose a model in which MOV10 alone unwinds secondary structures containing MREs and thus facilitates AGO2 binding whereas in association with FMRP, the complex is stabilized and AGO2 association is blocked (Figure 7B).

Minor point:

Could the authors discuss these antagonistic effects of FMRP on the regulation of the DICER mRNA, on one hand by facilitating the miRNA machinery assembly but on the other hand protecting DICER mRNA, in collaboration with Mov10, by inhibiting its recognition by the Ago2 complex?

In FMRP-KO brain, DICER protein levels are decreased but the levels of miRNA remain intact. To explain this apparent contradiction, the authors propose a role of FMRP, and Mov10, on the local expression of DICER protein in dendrites, leading consequently to local downstream expression of many genes.

Reviewer #3: This paper investigates the functional interaction between FMRP, MOV10 and Dicer and their role in dendritic development.

The authors report that: 1. Dendritic branching of cultured hippocampal neurons from Mov10 heterozygous and Fmr1 KO is similarly reduced compared with WT neurons; 2. Dendritic spines in cortical neurons of Mov Het at DIV 14 have more mature spines than WT (differently than Fmr1 KO); 3. Cell soma is smaller in Mov Het hippocampal cultured neurons compared WT neurons. However, cell soma of Fmr1 KO was not different than WT neurons. 4. A global reduction of miRNAs associated with AGO in the absence of FMRP 5. reduced DICER expression in the absence of both FMRP and AGO, however no substantial change in the expression of miRNAs in Fmr1 KO; 6. Overexpression of DICER restores dendritic arborization in MOV10 heterozygous, but not in Fmr1 KO.

The authors conclude that the MOV10-Fmrp-AGO2 complex regulates DiCER expression, which in turn affects dendritic development.

This is a good paper exploring an interesting aspect related to the functional interaction between MOV10 and FMRP.

There are a few concerns that need to be addressed.

1. Methods are clearly described with useful details. At page 7 the differentiation methods used for N2A cells should be described.

2. Abnormal morphology of dendrites in cultured hippocampal neurons from Mov10 Het and Fmr1 KO mice. The authors confirmed their previously published results (Skariah et al., 2017). They should also mention other papers addressing the same issue and obtaining similar (Braun and Segal, 2000) and different results in cultured Fmr1 KO hippocampal neurons (Jacob and Doering, 2007). An increased branching of Fmr1 KO hippocampal neurons has been ascribed to the presence of astrocytes. Did the authors consider the contribution of glia?

3. Figure 1. Legend of figure 1 (see also results) indicates that data are from 56, 94 and 58 neurons. However, they do not indicate how many dishes in different cultures were used to gain these results. This is important to verify how reproducible are the data in different dishes, and more importantly in different cultures.

4. Density of dendritic spines, page 14 Why the authors examined cortical neurons in the brain instead of hippocampal neurons, considering that they want to compare in vivo and in vitro conditions? Spines could be examined in vitro as well. In addition, the region/layer of cortex examined should be indicated. They also measured the width of dendritic branches. Which part of neuron dendrite was used to measure these different parameters, width of dendrites and density and morphology of spines?

5. Reduced soma size of Mov10 Het cultured hippocampal neurons. I suggest moving this part before the ex-vivo analysis of dendritic spines, to complete the in vitro examination of hippocampal morphology first.

6. Other papers have addressed the question whether miRNAs are differentially expressed in the Fmr1KO mice. They should be cited and discussed considering that the authors found no change in the whole brain at PO. Example:

Zhang M, Li X, Xiao D, Lu T, Qin B, Zheng Z, Zhang Y, Liu Y, Yan T, Han X. Identification of differentially expressed microRNAs and their target genes in the hippocampal tissues of Fmr1 knockout mice. Am J Transl Res. 2020 Mar 15;12(3):813-824. PMID: 32269714; PMCID: PMC7137065.

Liu T, Wan RP, Tang LJ, Liu SJ, Li HJ, Zhao QH, Liao WP, Sun XF, Yi YH, Long YS. A MicroRNA Profile in Fmr1 Knockout Mice Reveals MicroRNA Expression Alterations with Possible Roles in Fragile X Syndrome. Mol Neurobiol. 2015;51(3):1053-63. doi: 10.1007/s12035-014-8770-1. Epub 2014 Jun 7. PMID: 24906954.

7. References are not always properly cited (ex. Bolduc et al., 2008 does not refer to abnormal dendritic spines in Drosophila, but excessive protein synthesis; similarly, Kelleher and Bear, 2008 and Liu-Yesucevitz refer to mRNA translation rather than spine/dendritic dysmorphogenesis). Similarly, Batish et al., 2012 is a study addressing the mRNA travel along dendrites as single particle rather than complexes of RBPs and Contractor et al., 2015 does not address the dendritic maturation and neurite extension but the altered neuronal excitability in FXS.

8. The authors suggest that DICER reduction in Fmr1 KO and Mov10 Het may affect miRNAs locally instead than leading to a global reduction of miRNA production. The authors should discuss the opposite effect on dendritic spines in Mov10 and Fmr1 KO.

Darnell 2011 is cited twice.

6. PLOS authors have the option to publish the peer review history of their article (what does this mean?). If published, this will include your full peer review and any attached files.

Reviewer #1: No

Reviewer #2: No

Reviewer #3: No

---

## [Author Response · Author response to Decision Letter 0]

31 Aug 2021

Response to Reviewers

Thank you for this opportunity to fix mistakes and make this a stronger manuscript. 

Reviewer #1: 

My biggest concern is the lack of consistency across different experiments that make it hard to draw some of the conclusions. For example, the authors suggest that Dicer expression is only reduced in FMR1 KO brain but not MOV10 Het KO, but the brains they used were P2 while the spine defects of the MOV10 KO mice in Fig. 2 were observed at P14 (which match the 14 DIV neurons in Fig. 1). 

1.Is it possible that MOV10 deficiency in the Het KO reduces Dicer expression at P14, which can potentially explain the spine and dendrite phenotypes? 

To address this question, we performed Dicer western blots on P14 whole brains from WT and Mov10 Het and found no difference (see Supplementary Fig 3B). In addition, we performed Dicer immunostain on DIV14 cultured hippocampal neurons from WT and Mov10 Het and found no difference in overall intensity in these neurons. Thus, we hypothesize that DICER expression is controlled locally in the dendrite and spine. Unfortunately, more precise measurements of DICER localization in dendrites and spines are beyond the scope of this study. 

2.Likewise, Fig. 1 examines dendrite defects in cultured hippocampal neurons while Fig. 2 examines spine defects in cortical neuron in vivo using Golgi stain. It would be much better if they correlate dendrite and spine defects in the same group of neurons (either hippocampal or cortical, in vitro or in vivo). 

To expand our study of the cultured hippocampal neurons in Figure 1, we examined spines in the somatosensory cortex by Golgi staining. To address the reviewer’s concern, we re-examined the Golgi-stained hippocampal regions in those brain sections but unfortunately, it was over-saturated. 

3. Furthermore, the authors claim that the spine defects in MOV10 KO are different from FMRP KO, but the spines from MOV10 KO are at P14 while the spine defects in FMR1 KO mice are implicated from studies by other labs done at different ages (4- wk old and adult, p.14). Without side-by-side comparison in the same experiments, same age and same group of neurons it Is hard to draw this conclusion.

We appreciated the reviewer’s suggestion and analyzed the brains from P14 Fmr1 ko mice and found that they were similar to WT and that both were distinctly different from the Mov10 Het. We now include the measurements from the age-matched Fmr1 ko mice in Figure 3C. 

4.In Fig. 6C the MOV10 KO are N2A cells and FMR1 KD is done in 293T cells. They should not be put together in the same graph for comparison. 

Thank you for pointing this out: we have now made 2 figures in Figure 6 (C and D).

Other comments:

5.Fig. 1: how many independent experiments have been performed? 

Neuron culture was performed on at least 3 litters from each genotype and the N was compiled from the three experiments. We have now added this information to the figure 1 legend and in the methods. 

6.Fig. 2C: is there statistical significance between genotypes for mature spines and immature spines? 

We performed ANOVA analysis on WT mature vs. Mov10 Het mature and did not find a significant difference between these genotypes. 

7.Fig. 4B: does MOV10 KO brain show changes in the amount of Ago2-associated miRNAs compared to WT? And what do the two lines in the plot refer to? 

Unfortunately, we have not evaluated the MOV10 ko brain animal because we are still developing that animal. The solid black line is the best fit of the line for a perfect correlation and the dashed blue line is the actual fit of the data. We have added this information to the Figure 4 legend. 

8.Fig. 6D upper panel: Does the green signal represent MYC-Dicer OE cells? It should be stated clearly in the figure. The authors should also show the images for all three experimental conditions that match the quantification in the lower panel.

Yes, the green signal represents immunoreactivity to MYC epitope, which we have now added to the figure. We have created a new Figure 7 showing all of the experimental conditions with Dicer over-expression in both Neuro2A and in neurons. 

9.Fig. 6D: Top left image of the upper panel indicates WT + Dicer OE but there is no such condition in the histogram of the lower panel. Does the blue bar represent WT + Dicer OE? 

 We have now included WT + Dicer OE in the new Figure 7A

10.Fig. 6E: why the bars are broken?

We were trying to plot all of the data on one graph. In response to the reviewer, we replotted the data on the same scale for the Y axis (Figure 6E).

11.Fig. 6F: representative images of the MOV10 KO and FMR1 KO neurons with or without Dicer OE should be shown. 

We have now added representative images in Figure 7B and C. 

12.Fig. 6F: if Dicer OE can reverse the dendrite phenotypes of MOV10 Het hippocampal neurons, the authors need to examine whether Dicer expression is indeed reduced in the Het neurons compared to WT. 

We examined DICER expression in WT and MOV10 Het brain at P14 (Supplemental Fig 3B) and performed DICER immunostain on DIV14 cultured hippocampal neurons from WT and Mov10 Het and found no difference in intensity in these neurons (data not shown). Thus, we hypothesize that DICER expression is controlled locally in the dendrite and spine, which is not visualizable by any of these methods. 

13.Does OE Dicer restore the reduction of soma size in MOV10 Het neurons? 

We did this reanalysis but did not obtain a large enough sample size of transfected neurons to reject the null hypothesis.

14.P.3, second paragraph: the sentence is too long and should be divided into separate sentences.

We have done this. 

15.I think dendrite width is seldom measured. The authors should discuss the significance of a change in dendrite width observed in their study. 

This is an interesting question. We did this experiment because of the effect of MOV10 loss inNeuro2A on neurite width (Supplemental fig 1). The width reflects the maturation process as we now add the following description on the top of pg.17:

 “Recent work shows that dendritic widths may be shaped by intracellular transport and forces from the cytoskeleton and the area proportionality accords with a requirement for microtubules to transport materials and nutrients for dendrite tip growth (Liao et al., 2021). Thus, reduced MOV10 levels likely perturb dendrite formation because MOV10 binds cytoskeletal mRNAs (Skariah et al, 2017).”

16.P.19: Kennedy et al. 2020 is not found in the reference list

We have fixed this. 

17.P.23: “suggest a mechanism for regulating local DICER expression when MOV10 and FMRP are present”. I don’t think the authors have provided evidence showing a local dendritic (but not global) reduction of DICER. 

 The reviewer is correct. This is our proposed model. 

18.P.23: “there is also a large number of RNAs that are unique to FMRP and MOV10 and it is likely that misregulation of these mRNAs cause the unique spine features.” As mentioned in previous comment, it is not appropriate to conclude “unique spine features” from the available data because the findings on MOV10 and FMR1 KO were not done side-by-side and the ages or even populations of neurons examined are different. 

 We have now done this and provided the analysis in Figure 3C. 

Reviewer #2: 

Major points:

1.Figure 1 A: images and in particular the morphology of the selected Mov10 Het neuron do not reflect the quantitative characterization presented in B. At least 3 images of representative neurons from each condition should be presented.

 We have added these representative images to Figure 1.

2.Figure1B-E: Statistical tests should be detailed for each experiments. In particular, the number of different neuronal cultures should be specified. 

Fig. 1: how many independent experiments have been performed? 

We prepared neurons from 3 independent litters of each genotype and thus, 3 independent cultures. The total number, N, was compiled from the three biological experiments. We have now added this information to the Figure 1 legend and in the methods. 

3.Figure 3B and Supplemental Fig 2: the number of different neuronal cultures used for statistical analysis should be specified. Could the individual measurements of soma size area be provided? 

We used the neuron cultures described in Figure 1 for soma size analysis and have now included this information as Supplemental table 2.

Major points:

4.Figure 4 A: It seems that there are inconsistencies between the legend of the figure and the text. Is it GO enrichment for FMRP bound mRNA (legend) or differentially Ago2-associated mRNA in WT vs Fmr1-KO (text)? Were they identified by iCLIP or eCLIP? Where is the list available? The text refers to Kenny et al., 2020 and the references section mentions Kenny et al., 2019. 

 Thank you for pointing these issues out. The GO enrichment is for the 29 shared brain MOV10- and FMRP- and AGO-CLIP targets, as we now describe in the Figure 4 legend and in the Results. The MOV10 iCLIP list was published in Skariah et al 2017, the AGO2 eCLIP list was published in Kenny et al 2020 (not 2019, which we have now corrected) and the FMRP iCLIP list was published by Darnell 2011. 

5.Figure 4 B/ Sup table 1: legends are not accurate: what are full line, dashed line? 

We now describe the full and dashed lines in the Figure 4 legend.

The Supplementary Table 1.1 is of the significantly changed AGO2-bound miRNAs. 

6.How statistical analysis were performed?

 They were performed by ECLIPSE: They set a significance threshold of -log10(P-value) ≥ 3 and a log2 fold change ≥ 3, which is now added to the eCLIP methods. 

7.To rule out any indirect effect, the level of the Ago2 protein in WT vs Fmr1-KO P0 brains should be checked. 

We agree: the AGO2 levels are the same in Fmr1 knockout and WT brains. This was published in Supplemental Figure S6 of Kenny 2020. We now state this on the bottom of page 18.

8.The Dicer mRNA being a target of FMRP, MOV10 and Ago2, the authors analyzed the expression of the DICER protein and showed that DICER protein expression is decreased in Mov10-KO N2A cells or P2 Fmr1-KO brain extracts compare to WT (Figure 5A-B).

Major points:

To fully support the hypothesis that FMRP and MOV10 modulate the Ago2-miRNA mediated regulation of the Dicer mRNA, additional experiments are required:

What are the levels of Dicer mRNA in these different models? 

Dicer 1 mRNA levels are likely to be unchanged because Dicer1 mRNA is not among the mRNAs changed in the brains of Fmr1 ko mice compared to WT (Korb et al., 2017; Darnell et al 2011) nor in the MOV10 knockout Neuro2A cell line (Skariah et al., 2017). We believe regulation is at the level of translation. We address this on pg. 20, line 399

9.What is the impact of the absence of FMRP on the association of the Dicer mRNA with Mov10? and conversely?

Based on our earlier work showing that FMRP facilitates loading of mRNAs onto MOV10 (Kenny et al, 2014), we would predict that in the absence of FMRP, MOV10 binds less Dicer mRNA. We look forward to exploring this question when we have the Mov10 ko mouse in-hand. 

10.What is the impact of the absence of FMRP or Mov10 on the association of the Dicer mRNA with Ago2? 

We examined AGO-association with Dicer1 in the absence of FMRP in the Kenny 2020 study and found that it was reduced in the absence of FMRP but not significantly. 

11.What is the purpose of Figure 5C here? 

 We have moved this figure to the supplementary data. It is S3. 

12.As MOV10 and FMRP bind murine Dicer mRNA and human DICER mRNA respectively in the 3’UTR, the authors used a DICER1 3’UTR luciferase construct to dissect the potential role of FMRP and MOV10 in the modulation of DICER1 via the 3’UTR. They showed a decreased luciferase activity in the absence of FMRP or MOV10 proteins (Figure 6C).

Major points:

13.Figure 6: Both legend and Material and Methods sections poorly explain the luciferase assay and in particular how long and short constructs participate in the calculation of the Relative Luciferase Actvity (Could the authors explain: “Mov10 KO and Fmr1 KD samples were normalized to WT Dicer long 3’UTR and Dicer short 3’UTR values were subtracted for final graph”?) 

We apologize for not clearly stating our methodology: Dicer short 3’UTR contains the luciferase reporter only and allows us to examine the effect of FMRP or MOV10 loss on expression of the luciferase coding sequence itself. This is relevant as others have reported that FMRP binds luciferase coding sequence and affects expression (Chen et al. Mol Cell 2014). Dicer long 3’UTR allows us to determine the effects of the absence of MOV10 and FMRP on the entire 3’UTR. The final results had the background expression of Dicer short 3’UTR luciferase vector subtracted from the long results to account for effects on the reporter itself. Finally, we normalized to the WT expression of the Dicer luciferase vector which was set to 1. We have now added a clear description to the Methods. 

14. Many steps in mRNA biogenesis could affect the reporter protein expression. qPCR to evaluate the expression of the reporter mRNA in the different conditions would strengthen the authors hypothesis. 

Thank you for this suggestion. Our goal here was to report the effect on reporter expression in the absence of both MOV10 and FMRP. We do think the reduction is at the level of RNA transcription or degradation. 

15.By CLIP-seq data comparison and miRNA recognition elements (MRE) analysis, the authors determined potential MSE the in close proximity of FMRP and MOV10 binding sites in the DICER 3’UTR (Sup Figure 5).

16. Minor points:

Are mouse and human Dicer mRNA 3’UTR (FMRP binding sites, Mov10 binding sites, predicted MSE ) conserved? 

Thank you for asking. Yes! They are conserved and we have included this information in Supplemental figure 5C

17.Sup Figure 5: B and C should be harmonized to facilitate the reading. 

We modified Supplemental Figure 5 B and C

18.Major points:

1. Figure 6 D: Incomplete legend

 We have added information about how the analysis was performed.

2. Figure 6 E: Inconsistencies between text and legend.

 We have revised the figure legend to reflect the constructs introduced.

3. Figure 6 F left:

 Are Mov10 Het cells cultured hippocampal neurons? 7 DIV? 14 DIV?

 Yes, they are DIV7 in Figure 7

19.Were analyzed neurons detected for MYC-DICER expression? What is the level of over-expression in these neurons? 

Yes. We do not know the level of over-expression because we did not have enough material to immunoblot for DICER

20.Is heterologous expression homogenous?

We looked by immunostain but it was difficult to tell. It is probably heterogeneous because it is a transfection experiment 

21.How does the over-expression of DICER impact dendritic arborization in WT neurons? 

We did not examine WT neurons because we did not observe a difference when Dicer was over-expressed in Neuro2A (Fig 7A)

22.Can the authors comment the difference in the numbers of intersections between WT (Figure 1), Mov10 Het over- expressing negative CT and Mov10 Het over-expressing DICER (Figure 6F)? 

We believe that the differences are due to a differential experimental treatment and visualization. For example, Figure 1 was Map2 staining to maximize visualization of dendrites of DIV 14 cultures while Figure 7 was DIV 7 cultures along with exposure to Lipofectamine and plasmid DNA followed by immunostaining

23.Considering the cooperative model proposed by the authors, what is the effects of FMRP over-expression on the arborization of Mov10 Het neurons? 

That is in interesting question. We have not tested FMRP over-expression; however, despite working together to regulate a subset of mRNAs, MOV10 and FMRP each regulate a large number of independent mRNAs. 

24.Figure 6F right:

o Same questions as above regarding the control of MYC-DICER over-expression in analyzed neurons.Why is the negative control for FMR1-KO neurons different from the one used for Mov10 Het neurons?

 The negative controls were both empty expression vectors except for the tags. 

25.Can the authors comment the difference in the numbers of intersections between WT (Figure 1), Fmr1-KO over-expressing negative CT and Fmr1-KO t over-expressing DICER (Figure 6F)? 

Similar to what we wrote before, we believe that the differences are due to different experimental treatments, culture times and visualization. For example, Figure 1 was Map2 staining to maximize visualization of dendrites of DIV 14 cultures while Figure 7 was DIV 7 cultures along with exposure to Lipofectamine and plasmid DNA followed by immunostain. 

26. To conclude, the authors discuss the potential role of FMRP in facilitating the loading of miRNA on AGO2 (Figure 7A). In parallel, they propose a model in which MOV10 alone unwinds secondary structures containing MREs and thus facilitates AGO2 binding whereas in association with FMRP, the complex is stabilized and AGO2 association is blocked (Figure 7B). Minor point:

Could the authors discuss these antagonistic effects of FMRP on the regulation of the DICER mRNA, on one hand by facilitating the miRNA machinery assembly but on the other hand protecting DICER mRNA, in collaboration with Mov10, by inhibiting its recognition by the Ago2 complex? 

Thank you for this thought-provoking question. We believe that FMRP, like most RNA binding proteins, has many different functions in the cell, based on its phosphorylation state, its binding partners and its location in the cell. We might envision that in the cell body, DICER translation is facilitated by the association of FMRP and MOV10 with its mRNA. When the DICER-AGO-MOV10-FMRP complex is transported in the dendrite to the synapse, it awaits stimulation which activates DICER to produce miRNAs locally, which associate with AGO2 through its association with FMRP. Presumably translation is occurring of FMRP-MOV10 bound mRNAs because this complex blocks AGO association. However, translation must eventually stop. Perhaps methylation of the FMRP’s RGG box releases the FMRP-MOV10 complex from the 3’UTR of synaptically localized mRNAs and AGO now associates with the mRNAs to block translation. We have now added this paragraph to the Discussion. 

Reviewer #3: 

This is a good paper exploring an interesting aspect related to the functional interaction between MOV10 and FMRP. There are a few concerns that need to be addressed.

1. Methods are clearly described with useful details. At page 7 the differentiation methods used for N2A cells should be described.

 We added the specific details for differentiation to the Methods. 

2. Abnormal morphology of dendrites in cultured hippocampal neurons from Mov10 Het and Fmr1 KO mice. The authors confirmed their previously published results (Skariah et al., 2017). They should also mention other papers addressing the same issue and obtaining similar (Braun and Segal, 2000) and different results in cultured Fmr1 KO hippocampal neurons (Jacob and Doering, 2007). Did the authors consider the contribution of glia?.

We appreciate the reviewer pointing out these references. We have now added them to the manuscript. We did consider the role of glia. We do not culture on an astrocyte feeder layer, like was often done in the past. This may be why we see defects in the Fmr1 ko neurons. Others are now culturing without feeders and see a result similar to ours, i.e., impaired arborization of Fmr1 ko neurons, which we cite. 

3. Figure 1. Legend of figure 1 (see also results) indicates that data are from 56, 94 and 58 neurons. However, they do not indicate how many dishes in different cultures were used to gain these results. This is important to verify how reproducible are the data in different dishes, and more importantly in different cultures. 

We performed 3 independent experiments from at least 3 liters per genotype. N was compiled from the three experiments. We have now added this information to the figure 1 legend and in the methods. Neurons were cultured in 24 well plates from 3 litters from each genotype. 

4. Density of dendritic spines, page 14 Why the authors examined cortical neurons in the brain instead of hippocampal neurons, considering that they want to compare in vivo and in vitro conditions? Spines could be examined in vitro as well. In addition, the region/layer of cortex examined should be indicated. They also measured the width of dendritic branches. Which part of neuron dendrite was used to measure these different parameters, width of dendrites and density and morphology of spines? 

To expand our study, we examined spines in the somatosensory cortex in vivo, which we have now indicated in the methods, while examining dendrite morphology in hippocampal culture in vitro, as others have done (Galvez and Greenough, 2005). To address the reviewer’s concern, we re-examined the Golgi-stained hippocampal regions but unfortunately, it was over-saturated. 

With regard to the second question, for the width measurements, it was the width of the primary branches and for the density and morphology of spines it included a 10 micron region from each level of dendrite. We have also added this to the methods 

5. Reduced soma size of Mov10 Het cultured hippocampal neurons. I suggest moving this part before the ex-vivo analysis of dendritic spines, to complete the in vitro examination of hippocampal morphology first. 

 Thank you for this suggestion. We have done this. 

6. Other papers have addressed the question whether miRNAs are differentially expressed in the Fmr1KO mice. They should be cited and discussed considering that the authors found no change in the whole brain at PO. Example:

Zhang M, Li X, Xiao D, Lu T, Qin B, Zheng Z, Zhang Y, Liu Y, Yan T, Han X. Identification of differentially expressed microRNAs and their target genes in the hippocampal tissues of Fmr1 knockout mice. Am J Transl Res. 2020 Mar 15;12(3):813-824.

PMID: 32269714; PMCID: PMC7137065.Liu T, Wan RP, Tang LJ, Liu SJ, Li HJ, Zhao QH, Liao WP, Sun XF, Yi YH, Long YS. A MicroRNA Profile in Fmr1 Knockout Mice Reveals MicroRNA Expression Alterations with Possible Roles in Fragile X Syndrome. Mol Neurobiol. 2015;51(3):1053-63. doi: 10.1007/s12035-014-8770-1. Epub 2014 Jun 7. PMID: 24906954.

 Thank you for this suggestion. We have added this information to our manuscript. 

7. References are not always properly cited (ex. Bolduc et al., 2008 does not refer to abnormal dendritic spines in Drosophila, but excessive protein synthesis; similarly, Kelleher and Bear, 2008 and Liu-Yesucevitz refer to mRNA translation rather than spine/dendritic dysmorphogenesis). Similarly, Batish et al., 2012 is a study addressing the mRNA travel along dendrites as single particle rather than complexes of RBPs and Contractor et al., 2015 does not address the dendritic maturation and neurite extension but the altered neuronal excitability in FXS. Sure.

 We apologize for these errors and appreciate the opportunity to fix them. 

8. The authors suggest that DICER reduction in Fmr1 KO and Mov10 Het may affect miRNAs locally instead than leading to a global reduction of miRNA production. The authors should discuss the opposite effect on dendritic spines in Mov10 and Fmr1 KO. 

Thinking about the reviewer’s question, it would seem like FMRP and MOV10 are working together in dendrite formation in cultured neurons but have discrete functions at P14 spines. We would hypothesize that the mRNAs that regulate dendritic arborization are distinct in time and perhaps place in the brain than those regulating spine morphology. 

9.Darnell 2011 is cited twice.

Thank you for pointing this out. We have removed this.

---

## [Decision Letter · Decision Letter 1]

12 Oct 2021

PONE-D-21-14401R1FMRP and MOV10 regulate Dicer1 expression and dendrite developmentPLOS ONE

Dear Dr. Ceman,

Thank you for submitting your manuscript to PLOS ONE. After careful consideration, we feel that it has merit but does not fully meet PLOS ONE’s publication criteria as it currently stands. Therefore, we invite you to submit a revised version of the manuscript that addresses the points raised during the review process.

Please consider the criticism of the reviewer 2 to be able to improve your manuscript

Please submit your revised manuscript by November 1st, 2021 If you will need more time than this to complete your revisions, please reply to this message or contact the journal office at plosone@plos.org. Please include the following items when submitting your revised manuscript:A rebuttal letter that responds to each point raised by the academic editor and reviewer(s). You should upload this letter as a separate file labeled 'Response to Reviewers'.A marked-up copy of your manuscript that highlights changes made to the original version. You should upload this as a separate file labeled 'Revised Manuscript with Track Changes'.An unmarked version of your revised paper without tracked changes. You should upload this as a separate file labeled 'Manuscript'.If applicable, we recommend that you deposit your laboratory protocols in protocols.io to enhance the reproducibility of your results. Protocols.io assigns your protocol its own identifier (DOI) so that it can be cited independently in the future. For instructions see: https://journals.plos.org/plosone/s/submission-guidelines#loc-laboratory-protocols. Additionally, PLOS ONE offers an option for publishing peer-reviewed Lab Protocol articles, which describe protocols hosted on protocols.io. Read more information on sharing protocols at https://plos.org/protocols?utm_medium=editorial-email&utm_source=authorletters&utm_campaign=protocols.

We look forward to receiving your revised manuscript.

Kind regards,

Barbara Bardoni

Academic Editor

PLOS ONE

Journal Requirements:

Additional Editor Comments (if provided):

Please answer to the minor criticisms of reviewer 2.

Reviewers' comments:

Reviewer's Responses to Questions

**Comments to the Author**

1. If the authors have adequately addressed your comments raised in a previous round of review and you feel that this manuscript is now acceptable for publication, you may indicate that here to bypass the “Comments to the Author” section, enter your conflict of interest statement in the “Confidential to Editor” section, and submit your "Accept" recommendation.

Reviewer #1: All comments have been addressed

Reviewer #2: (No Response)

Reviewer #3: All comments have been addressed

2. Is the manuscript technically sound, and do the data support the conclusions?

Reviewer #1: (No Response)

Reviewer #2: Partly

Reviewer #3: Yes

3. Has the statistical analysis been performed appropriately and rigorously? 

Reviewer #1: (No Response)

Reviewer #2: No

Reviewer #3: Yes

4. Have the authors made all data underlying the findings in their manuscript fully available?

Reviewer #1: (No Response)

Reviewer #2: Yes

Reviewer #3: Yes

5. Is the manuscript presented in an intelligible fashion and written in standard English?

Reviewer #1: (No Response)

Reviewer #2: Yes

Reviewer #3: Yes

6. Review Comments to the Author

Reviewer #1: (No Response)

Reviewer #2: The authors greatly improved the manuscript, detailing some experimental procedures, adding complementary controls and reinforcing the discussion section. However, it still remains few points that need to be clarified:

1- Reduced expression of Mov10 leads to smaller soma size

New figure 2, supplementary figure 2 and Supplementary table 1-2

There is a discrepancy between the individual measurements of soma size area provided in the Supplementary table 1-2 and the figures.

Figure 2: Some extreme values do not appear on the scatter plot. Is it due to an error in the graph scaling or were the values discarded as outliers? In this latter case, the mean and the sem should be revised.

According to the scatter plot, the values in the WT condition in Figure 2 and Supplementary Figure 2 are different (different mean and/or sem). However, the Supplementary table 1-2 presents a single serie of WT values.

Could the statistical options for the Student t-test be detailed? Is there a Welsh correction? Is the test one or two tailed?

2- Global reduction of AGO2-associated miRNAs in the absence of FMRP

Thank you to the authors for these further details. However, I still have a question:

Lane 358-359: Using this method, we found that the significantly enriched peaks fell within 279 miRNAs (p < .001, Fig 4A, Supplementary Table 1-1).

According to the supplementary table "Ago2 eCLIP-miRNA Table" and statistical analysis provided for IP1 WT , 531 hits present an IP1 vs Input1 -log10(P-value) ≥ 3, including 523 hits with an IP1 vs. Input1 log2 Fold Change ≥ 3. How were the 279 miRNAs mentioned in the text selected? Could they be highlighted in the supplementary table?

3- Dicer1 3’UTR regulation by MOV10 and FMRP

Response to the reviewer: Finally, we normalized to the WT expression of the Dicer luciferase vector which was set to 1. We have now added a clear description to the Methods.

Thank you to the authors for these further details. Regarding these precisions, I still have a question: Figure 6 C, D, E and Supplementary Figure S5 A: Relative Luciferase Activity: If the values were normalized to the WT values, how a sem can be calculated for the WT conditions?

4- Additional point:

Student's t test is not recommended as a pots-hoc test after a significant one way anova test.

Reviewer #3: The authors have answered to all my concerns.

There are still a few typos.

1. Pag 14 Correct Fig 1E

2. Page 19 erase ?

7. PLOS authors have the option to publish the peer review history of their article (what does this mean?). If published, this will include your full peer review and any attached files.

Reviewer #1: No

Reviewer #2: No

Reviewer #3: No

---

## [Author Response · Author response to Decision Letter 1]

29 Oct 2021

Thank you for this opportunity to make corrections on the manuscript. We appreciate the opportunity to respond to the comments below. Our answers are in red. 

Reviewer #2: The authors greatly improved the manuscript, detailing some experimental procedures, adding complementary controls and reinforcing the discussion section. However, it still remains few points that need to be clarified:

1- Reduced expression of Mov10 leads to smaller soma size

New figure 2, supplementary figure 2 and Supplementary table 1-2

There is a discrepancy between the individual measurements of soma size area provided in the Supplementary table 1-2 and the figures.

Figure 2: Some extreme values do not appear on the scatter plot. Is it due to an error in the graph scaling or were the values discarded as outliers? In this latter case, the mean and the sem should be revised.

According to the scatter plot, the values in the WT condition in Figure 2 and Supplementary Figure 2 are different (different mean and/or sem). However, the Supplementary table 1-2 presents a single serie of WT values.

Could the statistical options for the Student t-test be detailed? Is there a Welsh correction? Is the test one or two tailed?

Response: We looked into this and it appears to be a scaling issue when the graph is generated in our graphing software, Graphpad Prism. The software is automatically optimizing the y-axis for all data points to create a more cohesive image when extreme values are detected. In the case of Figure 2, the software excluded three outlier values and for Supplementary Figure 2 it excluded two outlier values. Each soma measurement was performed from the neurons imaged and analyzed for dendrite morphology in Figure 1, therefore only a single series of values for each genotype analyzed. We included an early version of Supplementary Figure 2 which was used for our power analysis prior to measuring all the values. We regret not catching this sooner. We have updated Figure 2B and Supplementary Figure 2. For soma size measurements, we performed a one-tailed t-test with Welch's correction. We have now specified this information clearly for all figures where that test was performed. 

2- Global reduction of AGO2-associated miRNAs in the absence of FMRP

Thank you to the authors for these further details. However, I still have a question:

Lane 358-359: Using this method, we found that the significantly enriched peaks fell within 279 miRNAs (p < .001, Fig 4A, Supplementary Table 1-1).

According to the supplementary table "Ago2 eCLIP-miRNA Table" and statistical analysis provided for IP1 WT , 531 hits present an IP1 vs Input1 -log10(P-value) ≥ 3, including 523 hits with an IP1 vs. Input1 log2 Fold Change ≥ 3. How were the 279 miRNAs mentioned in the text selected? Could they be highlighted in the supplementary table?

Response: We thank the reviewer for this question. Briefly, we de-duplicated the genes in Supplementary Table 1-1 and meant to report the number of unique miRNAs. To get this value, we selected all of the hits with -log(P-value)>3 and deduplicated the gene name list. “279” was a mistake—we meant to write “261”. The names of the 261 significantly enriched miRNAs can be found on the second sheet of Supplementary Table 1-1 named “Sig. enriched miRNAs”. We corrected 279 enriched peaks to 261

3- Dicer1 3’UTR regulation by MOV10 and FMRP

Response to the reviewer: Finally, we normalized to the WT expression of the Dicer luciferase vector which was set to 1. We have now added a clear description to the Methods.

Thank you to the authors for these further details. Regarding these precisions, I still have a question: Figure 6 C, D, E and Supplementary Figure S5 A: Relative Luciferase Activity: If the values were normalized to the WT values, how a sem can be calculated for the WT conditions?

Response: Thank you for pointing out our mistake: we should not have had an error bar denoting the sem for the WT value. The error bar corresponded to the pre normalization value and was incorrectly copied when the graph was remade. We have now corrected this on all relevant figures.

4- Additional point:

Student's t test is not recommended as a pots-hoc test after a significant one way anova test.

Response: We have removed the incorrect information from the manuscript. 

Reviewer #3: The authors have answered to all my concerns.

There are still a few typos.

1. Pag 14 Correct Fig 1E

2. Page 19 erase ?

Response: We have addressed these.

---

## [Editor Report · Decision Letter 2]

2 Nov 2021

FMRP and MOV10 regulate Dicer1 expression and dendrite development

PONE-D-21-14401R2

Dear Dr. Ceman,

We’re pleased to inform you that your manuscript has been judged scientifically suitable for publication and will be formally accepted for publication once it meets all outstanding technical requirements.

Kind regards,

Barbara Bardoni

Academic Editor

PLOS ONE
---

## [Editor Report · Acceptance letter]

16 Nov 2021

PONE-D-21-14401R2 

FMRP and MOV10 regulate Dicer1 expression and dendrite development 

Dear Dr. Ceman:

I'm pleased to inform you that your manuscript has been deemed suitable for publication in PLOS ONE. Congratulations! Your manuscript is now with our production department. 

Kind regards, 

on behalf of

Dr. Barbara Bardoni 

Academic Editor

PLOS ONE